# Quality control for single-cell analysis of high-plex tissue profiles using CyLinter

Gregory J. Baker [1,2,3] ✉, Edward Novikov[1,2,4], Ziyuan Zhao[5], Tuulia Vallius [1,2], Janae A. Davis[6], Jia-Ren Lin [2], Jeremy L. Muhlich [2], Elizabeth A. Mittendorf[6,7,8], Sandro Santagata [1,2,3,9], Jennifer L. Guerriero [1,2,6,7,8] & Peter K. Sorger [1,2,3] ✉

Tumors are complex assemblies of cellular and acellular structures patterned on spatial scales from microns to centimeters. Study of these assemblies has advanced dramatically with the introduction of high-plex spatial profiling. Image-based profiling methods reveal the intensities and spatial distributions of 20–100 proteins at subcellular resolution in $10^3$–$10^7$ cells per specimen. Despite extensive work on methods for extracting single-cell data from these images, all tissue images contain artifacts such as folds, debris, antibody aggregates, optical aberrations and image processing errors that arise from imperfections in specimen preparation, data acquisition, image assembly and feature extraction. Here we show that these artifacts dramatically impact single-cell data analysis, obscuring meaningful biological interpretation. We describe an interactive quality control software tool, CyLinter, that identifies and removes data associated with imaging artifacts. CyLinter greatly improves single-cell analysis, especially for archival specimens sectioned many years before data collection, such as those from clinical trials.

Tissues are complex assemblies of many cell types whose proportions and properties are controlled by cell-intrinsic molecular programs and interactions with the tumor microenvironment. Recently developed highly multiplexed tissue imaging methods (for example, MxIF[1], CyCIF[2,3], CODEX[4], 4i[5], mIHC[6], MIBI[7], IBEX[8] and IMC[9]) have made it possible to collect single-cell data on 20–100 proteins and other biomolecules in preserved two-dimensional and three-dimensional tissue microenvironments[10–15]. Such data are powerful complements to those obtained using dissociative methods such as single-cell RNA sequencing (scRNA-seq)[10,14–16]. Imaging approaches compatible with formalin-fixed, paraffin-embedded (FFPE) specimens are particularly powerful because they can tap into large archives of human biopsy and resection specimens[10,15,17] and also assist in the study of mouse models of disease[18].

Generating single-cell data from high-plex images requires segmenting images to produce single-cell 'spatial feature tables' that are analogous to count tables in scRNA-seq[19]. In their simplest form, each row in a spatial feature table contains the X,Y coordinate of a cell (commonly the centroid of the nucleus) and integrated signal intensities for each protein marker[20]. Cell types (for example, cytotoxic T cells immunoreactive to CD45, CD3 and CD8 antibodies) are then inferred from these tables, and spatial analysis is performed to

[1]Ludwig Center for Cancer Research at Harvard, Harvard Medical School, Boston, MA, USA. [2]Laboratory of Systems Pharmacology, Program in Therapeutic Science, Harvard Medical School, Boston, MA, USA. [3]Department of Systems Biology, Harvard Medical School, Boston, MA, USA. [4]Harvard John A. Paulson School of Engineering and Applied Sciences, Harvard University, Cambridge, MA, USA. [5]Systems, Synthetic, and Quantitative Biology Program, Harvard University, Cambridge, MA, USA. [6]Breast Tumor Immunology Laboratory, Dana-Farber Cancer Institute, Boston, MA, USA. [7]Breast Oncology Program, Dana-Farber/Brigham and Women's Cancer Center, Boston, MA, USA. [8]Division of Breast Surgery, Department of Surgery, Brigham and Women's Hospital, Boston, MA, USA. [9]Department of Pathology, Brigham and Women's Hospital, Harvard Medical School, Boston, MA, USA. ✉e-mail: gregory_baker2@hms.harvard.edu; peter_sorger@hms.harvard.edu

identify recurrent short- and long-range interactions associated with an independent variable such as drug response, disease progression or genetic perturbation.

High-plex spatial analysis has been performed using both tissue microarrays (TMAs), which comprise 0.3–1.5-mm-diameter 'cores' (~$10^4$ cells) from dozens to hundreds of clinical specimens arrayed on a slide, and whole-slide imaging, which can involve areas of tissue as large as 4–6 cm$^2$ (~$10^7$ cells). Whole-slide imaging is a United States Food and Drug Administration (FDA) requirement for clinical diagnosis[21], research and spatial power[15], but TMAs are nonetheless in widespread use. In this Article, we show that accurate processing of images from both types of specimen is complicated by the presence of imaging artifacts such as tissue folds, slide debris (for example, lint) and staining artifacts. The problem impacts all data we have examined but is particularly acute with specimens stored for extended periods on glass slides. In our study, this scenario is represented by 25 specimens from the TOPACIO clinical trial of 'Niraparib in Combination with Pembrolizumab in Patients with Triple-Negative Breast Cancer or Ovarian Cancer' (NCT02657889)[22], which was completed in 2021. We demonstrate the impact of artifacts on analysis of CyCIF images of TOPACIO tissue specimens and high-plex CyCIF, CODEX and mIHC datasets from several recently published studies. We then develop a human-in-the loop approach to remove single-cell data affected by microscopy artifacts using a software tool, CyLinter (code and documentation at https://lab-syspharm.github.io/cylinter/), that is integrated into the Python-based Napari image viewer[23]. We demonstrate that CyLinter can salvage otherwise uninterpretable multiplex imaging data, including those from the TOPACIO trial. Finally, we demonstrate progress on a deep learning (DL) model for automated artifact detection; libraries of artifacts identified using CyClinter represent ideal training data for such a model. Our findings suggest that artifact removal should be a standard component of processing pipelines for image-based spatial profiling data.

## Results

### Identifying recurrent image artifacts in multiplex IF images

To categorize imperfections and image artifacts commonly encountered in high-plex images of tissue, we examined seven datasets collected using three different imaging methods: (1) 20-plex CyCIF images of 25 triple-negative breast cancer (TNBC) specimens collected from TOPACIO clinical trial patients; (2) a 22-plex CyCIF image of a colorectal cancer (CRC) resection[15]; (3) a 21-plex CyCIF TMA dataset comprising 123 healthy and cancerous tissue cores[19]; (4) two 16-plex CODEX images of a single head and neck squamous cell carcinoma (HNSCC) specimen; (5) a 19-plex mIHC image of normal human tonsil[19]; (6) 59-plex and (7) 54-plex independent CODEX images of normal large intestine

(Supplementary Fig. 1a–g and Supplementary Table 1). Raw image tiles were processed using MCMICRO[19] to generate stitched and registered multi-tile image files and their associated single-cell spatial feature tables. Single-cell data were visualized as Uniform Manifold Approximation and Projection (UMAP)[24] embeddings clustered with HDBSCAN—an algorithm for hierarchical density-based clustering[25]. Images were also inspected by experienced microscopists and board-certified pathologists to identify imaging artifacts.

All specimens comprised 5-μm-thick tissue sections mounted on slides in the standard manner. This involves cutting FFPE blocks with a microtome and floating sections on water before capturing them on glass slides. Even in the hands of skilled histologists, this process can introduce folds in the tissue. We identified multiple instances of tissue folds in whole-slide and TMA specimens (Fig. 1a, Extended Data Fig. 1a and Online Supplementary Fig. 1a). Moreover, we found that cells within tissue folds gave rise to discrete clusters in UMAP feature space due to higher-than-average signals relative to unaffected regions of tissue (Fig. 1a,b and Extended Data Fig. 1a,b).

Bright antibody aggregates were common and also formed discrete clusters in UMAP space (Fig. 1c), as were debris in the shape of lint fibers and hair (Fig. 1d and Online Supplementary Fig. 1b). Despite having relatively low numbers of segmented cells, regions of necrotic tissue also exhibited high levels of background antibody labeling (Fig. 1e). Some specimens contained air bubbles probably introduced when coverslips were overlayed on specimens before imaging (Fig. 1f and Online Supplementary Fig. 1c). In principle, artifacts such as tissue folds and air bubbles can be reduced by skilled experimentalists, but access to the original tissue blocks and/or image reacquisition is required.

Additional artifacts were introduced at the point of imaging. These included out-of-focus image tiles due to sections not lying completely flat on the slide (Fig. 1g and Online Supplementary Fig. 1d), fluctuations in background intensity between image tiles (Fig. 1h) and miscellaneous aberrations that significantly increased signal intensities over image background (Fig. 1i) and generated discrete clusters in UMAP space (Fig. 1j and Extended Data Fig. 1c). In some cases, removal of artifacts revealed more subtle problems such as the presence of cells stained nonspecifically by all antibodies (for example, CODEX Dataset 6; Extended Data Fig. 1d,e). Errors were also observed in tile stitching (Fig. 1k) and registration (Fig. 1l); in some cases, these problems can be addressed by reprocessing the data, but oversaturation of nuclear stain used for stitching and registration may still limit the accuracy of reprocessed data.

Some artifacts were specific to cyclic imaging methods such as CyCIF[2,3], CODEX[4] and mIHC[6] that generate high-plex images through multiple rounds of lower-plex imaging followed by fluorophore

**Fig. 1 | Recurring artifacts in whole-slide immunofluorescence images of tissue and their effects on tissue-derived single-cell data. a**, Top: Dataset 6 (large intestine, CODEX, specimen 1) containing a tissue fold (ROI, dashed white outline) as seen in channels SOX9 (colormap) and Hoechst (gray). Bottom: UMAP embedding of 57-channel single-cell data from the image above colored by SOX9 intensity (top left), ROI inclusion (top right) and HDBSCAN cluster (bottom center). Cluster 1 cells are those affected by the fold. **b**, Channel z scores for HDBSCAN clusters in **a** demonstrating that cluster 1 cells are artificially bright for all markers. **c**, Left: antibody aggregate in the CD63 channel (colormap) of Dataset 3 (EMIT TMA, core 68, normal tonsil). Other channels shown for context. Right: UMAP embedding of 20-channel single-cell data from the image shown at left colored by CD63 intensity (top) and ROI inclusion (bottom). **d**, Autofluorescent fiber in Dataset 1 (TOPACIO, specimen 128) as seen in channels 53BP1 (green) and Hoechst (gray). **e**, Necrosis in a region of tissue from Dataset 1 (TOPACIO, specimen 39) as seen in the CD3 channel (green). **f**, Coverslip air bubbles (green asterisks) in Dataset 1 (TOPACIO, specimen 48) as seen in the Hoechst channel (gray). **g**, Out-of-focus region of tissue in Dataset 1 (TOPACIO, specimen 55) as seen in the Hoechst channel (gray). **h**, Uneven tile illumination in Dataset 4 (HNSCC, CODEX, section 1) as seen in an empty Cy5

channel (green). AFU, arbitrary fluorescence units; s.d., standard deviation. **i**, Bottom: illumination aberration in the pCREB channel (colormap) of Dataset 3 (EMIT TMA, core 95, dedifferentiated liposarcoma) with superimposed nuclear segmentation outlines (translucent contours). Top: line plot demonstrating that artifactual per cell pCREB signals reach an order of magnitude above background. **j**, Top: field of view from Dataset 7 (large intestine, CODEX, specimen 2) showing five illumination aberrations (ROIs, dashed white outlines) in the CD3 channel (colormap). Bottom: UMAP embedding of 52-channel single-cell data from the image above colored by CD3 intensity (left) and ROI inclusion (right). **k**, Tile stitching errors in Dataset 5 (mIHC, normal human tonsil) as seen in the PD1 (green) channel. **l**, Cross-cycle image registration error in Dataset 3 (EMIT TMA, core 64, leiomyosarcoma) as demonstrated by the superimposition of cycle 1 Hoechst (gray) and cycle 9 pCREB (green) signals. **m**, Cross-cycle tissue movement in Dataset 1 (TOPACIO, specimen 80) as demonstrated by the superimposition of Hoechst signals from sequential imaging cycles: 1 (red), 2 (green) and 3 (blue). **n**, Progressive tissue loss in Dataset 3 (EMIT TMA, core 1, normal kidney cortex) across ten imaging cycles as observed in the Hoechst channel (gray). **o**, UMAP embedding of cells from Dataset 3 (EMIT TMA, core 1, normal kidney cortex) colored by stability.

dissociation or inactivation. For example, tissue movement (Fig. 1m) and tissue damage (Fig. 1n) caused cells present in early rounds of imaging to be lost at later cycles. These cells appear negative for all markers after movement or loss, confounding cell type assignment and leading to artifactual clusters in feature space (Fig. 1o). The extent of tissue loss varies between specimens and seems to arise during tissue dewaxing and antigen retrieval[26] owing to low tissue area (for example, the fine-needle biopsies from TOPACIO patients 70, 89, 95 and 96) and cellularity (for example, adipose tissue).

The origins of some artifacts remain unknown but probably arise from a combination of (1) preanalytical variables—generally defined as variables arising before specimen staining, (2) unwanted fluorescent objects (for example, lint and antibody aggregates) introduced during staining, imaging and washing steps, (3) errors in data acquisition and (4) the intrinsic properties of the tissue itself[27,28]. The TOPACIO specimens (Dataset 1) were the most severely affected by these artifacts, whereas the CRC specimen (Dataset 2), which had been freshly sectioned and carefully processed, was much less affected. However, only one slide was available from each TOPACIO patient, making repeat imaging impossible.

## Microscopy artifacts obscure single-cell analysis

Clustering Dataset 2 (CRC, CyCIF, ~9.8 × 10^5 total cells) with HDBSCAN yielded 22 clusters with 0.7% of cells remaining unclustered (Fig. 2a). Silhouette analysis[29] showed that four clusters (6, 15, 17 and 21) remained underclustered despite parameter tuning as indicated by negative

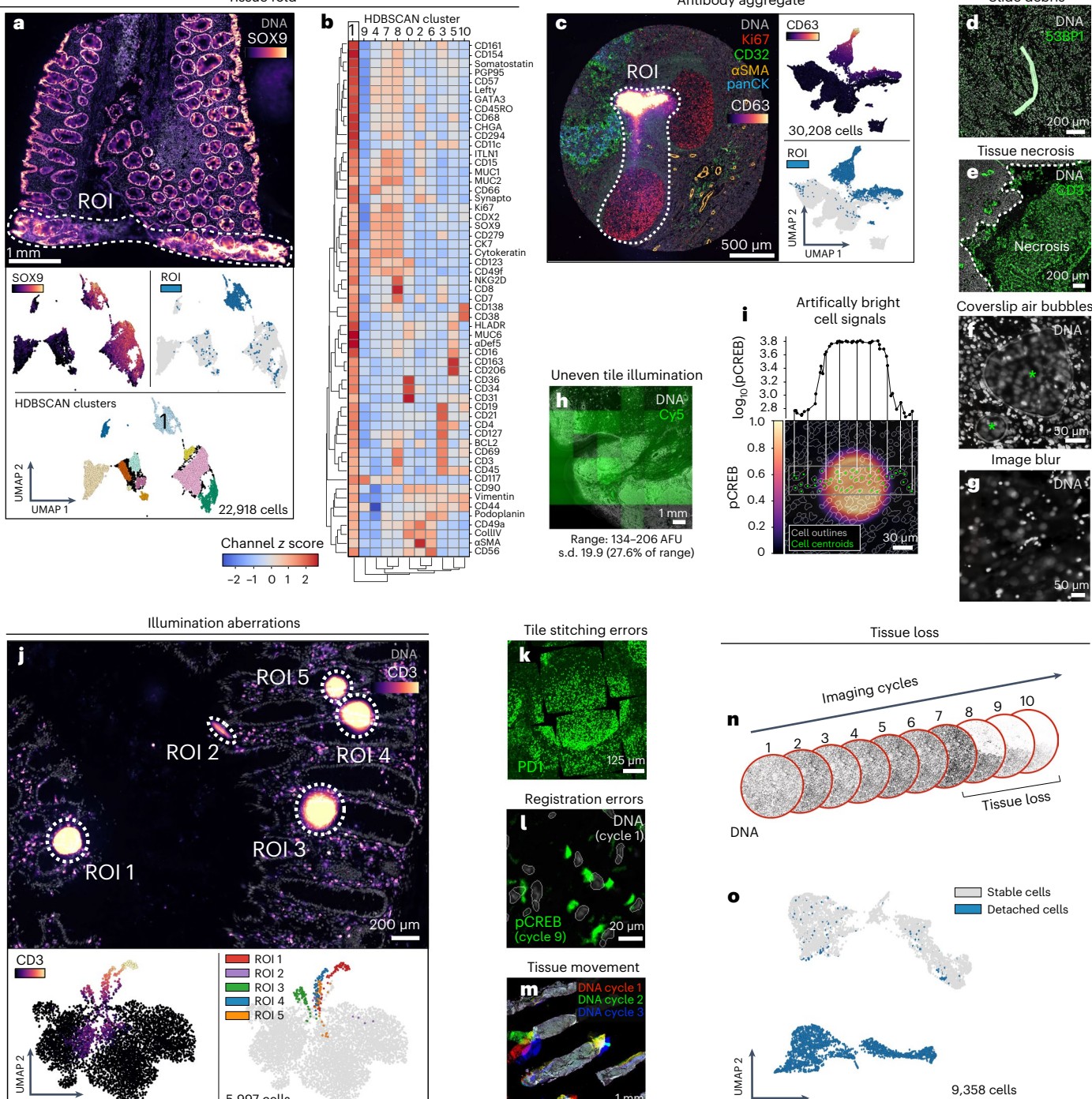

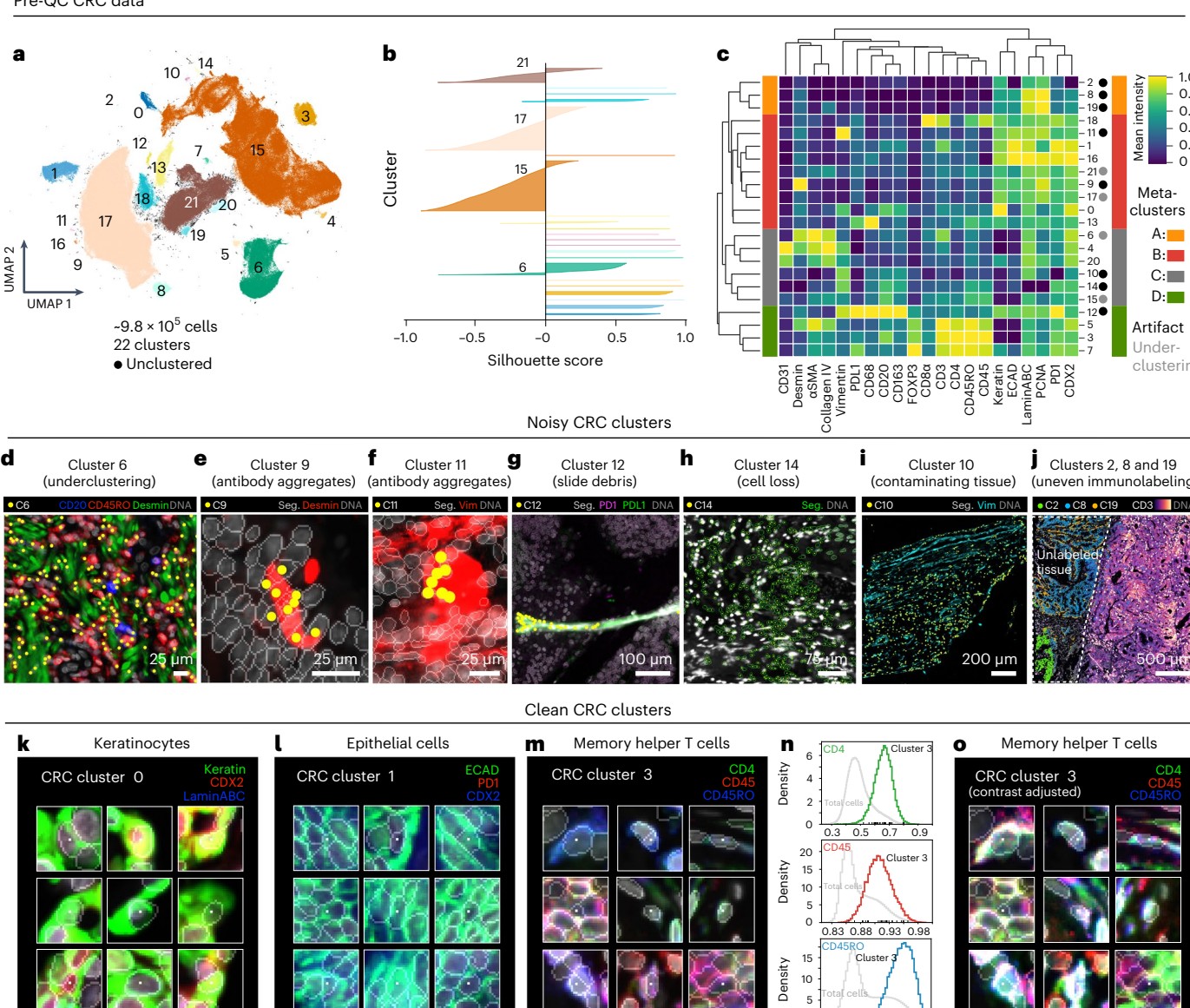

**Fig. 2 | Evaluation of pre-QC cell clustering results from Dataset 2 (CRC).**
**a**, UMAP embedding of CRC data showing ~9.8 × 10⁵ cells colored by HDBSCAN cluster. Black scatter points represent unclustered (ambiguous) cells.
**b**, Silhouette scores for CRC clusters shown in **a**. **c**, Mean signals of clustering cells in the CRC dataset normalized across clusters (row-wise). Four (4) meta-clusters defined by the heatmap dendrogram are highlighted. **d**, Cluster 6 cells (yellow points) in a region of the CRC image demonstrating the co-clustering of B cells (CD20, blue), memory T cells (CD45RO, red) and stromal cells (desmin, green). **e**, Anti-desmin antibody aggregates (red) in the CRC image. Yellow points highlight cluster 9 cells formed due to this artifact. **f**, Anti-vimentin antibody aggregates (red) in the CRC image. Yellow points highlight cluster 11 cells formed due to this artifact. **g**, Autofluorescent fiber in the CRC image as seen in channels PD1 (magenta) and PDL1 (green). Yellow points highlight cluster 12 cells formed due to this artifact. **h**, Cell loss in the CRC image as indicated by anucleate segmentation outlines (green). Yellow points highlight cluster 14 cells formed due to this artifact. **i**, Contaminating (noncolonic) tissue in the CRC image

immunoreactive to anti-vimentin antibodies (cyan) comprising CRC cluster 10 (yellow points). **j**, Region of tissue in the CRC image unexposed to antibodies during imaging cycle 3 leading to the formation of CRC clusters 2, 8 and 19 as observed in the CD3 channel (colormap). **k**–**m**, Top three most highly expressed markers (1, green; 2, red; 3, blue) for clusters 0 (keratinocytes, **k**), 1 (crypt-forming mucosal epithelial cells, **l**), and 3 (memory helper T cells, **m**). A single white pixel at the center of each image patch highlights the reference cell. Nuclear segmentation outlines (translucent white outlines) and Hoechst (gray) shown for reference. **n**, Density histograms showing the distribution of cluster 3 cells according to channels CD4 (green outline), CD45 (red outline) and CD45RO (blue outline) superimposed on distributions of total cells according to the same channels (gray outlines). Rugplots at the bottom of each histogram show where 25 cluster 3 cells (shown in Extended Data Fig. 2h) reside in each distribution. **o**, Cluster 3 cells shown in **m** after per channel and per image adjustment of signal intensity cutoffs to improve their homogeneity of appearance.

silhouette scores for some cells in those clusters (Fig. 2b). Agglomerative hierarchical clustering of HDBSCAN clusters based on mean marker intensities revealed four meta-clusters (Fig. 2c) corresponding to tumor (meta-clusters A and B), stromal (C) and immune (D) cell populations. To gain further insight into these HDBSCAN clusters, cells from each

were selected at random and organized into galleries of 20 × 20 μm (30 × 30 pixel) image patches centered on reference nuclei (Online Supplementary Fig. 2). To facilitate interpretation, only the three most highly expressed protein markers were shown per cluster (based on channel intensities normalized across clusters; Fig. 2c). Inspection

of these galleries showed that many clusters contained mixed cell types. For example, cluster 6 contained B cells, T cells and stromal cells (Fig. 2d). The formation of clusters 9 and 11 was driven by bright antibody aggregates in the desmin and vimentin channels (Fig. 2e,f), respectively, whereas contaminating lint fibers led to the formation of cluster 12 (Fig. 2g). Cell loss was evident in cluster 14 (Fig. 2h), and cluster 10 comprised a domain of vimentin-positive tissue of unknown origin (Fig. 2i). Three additional clusters (2, 8 and 19; Fig. 2j) were caused by a region of tissue unexposed to antibodies during imaging cycle 3 as evidenced by a sharp cutoff in immunolabeling in this area. We reasoned that this artifact was probably due to human error during the performance of a complex three-dimensional imaging experiment[15]. Clustering of Dataset 6 (CODEX, large intestine) also revealed clusters in which the expected separation of cell types was confounded by antibody aggregates, tissue folds and image blur (Extended Data Fig. 2a–f and Online Supplementary Fig. 3). We conclude that the presence of image artifacts, even in relatively unaffected specimens, can drive the formation of clusters that contain cells of different types.

Many other clusters in Dataset 2 (for example, 0, 1, 3, 7 and 16) contained few obvious artifacts. For example, cluster 0 comprised a phenotypically homogeneous group of keratinocytes (Fig. 2k), while cluster 1 represented normal crypt-forming epithelial cells (Fig. 2l). Cluster 3 consisted of CD4, CD45 and CD45RO$^+$ memory T cells distributed throughout the tissue (Extended Data Fig. 2g). Cells in this cluster appeared remarkably nonuniform (Fig. 2m and Extended Data Fig. 2h), despite their occupying a discrete region of the UMAP embedding (Fig. 2a) and having CD4, CD45 and CD45RO levels well above background (Fig. 2n). Protein expression among these cells was also well correlated ($R = 0.56$–$0.59$; Extended Data Fig. 2i), suggesting that cluster 3 encompassed a single cell population. Consistent with this conclusion, adjusting image intensity on a per-channel and per-cell basis resulted in their more uniform appearance (Fig. 2o and Extended Data Fig. 2j,k). Cells in cluster 7 ($T_{Reg}$ cells; Extended Data Fig. 2l) also formed a tight cluster (Fig. 2a) with good correlation in expression of CD4, CD45 and CD45RO ($R = 0.51$–$0.62$; Extended Data Fig. 2m) but weak correlation with FOXP3, the defining transcription factor for $T_{Reg}$ cells ($R = 0.13$–$0.31$; Extended Data Fig. 2n). We conclude that nonuniformity in the appearance of these cells probably arises from natural cell-to-cell variation in protein levels[30]—not simply dataset noise—but that multidimensional clustering correctly groups such cells into biologically meaningful subtypes. Thus, visual review must be performed with care and, ideally, in conjunction with data-driven approaches such as HDBSCAN.

Clustering Dataset 1 (25 TOPACIO specimens) gave rise to 492 HDBSCAN clusters with ~29% of cells remaining unclustered (Fig. 3a) and exhibiting no discernible spatial pattern in the underlying images (Extended Data Fig. 3a). Most clusters were associated with positive silhouette scores, indicating a good fit (Fig. 3b). While a few small clusters contained cells from a single tissue specimen (for example, cluster 75 with 418 cells and cluster 146 with 2,140 cells), most clusters (441/492)

contained cells from more than half of the 25 TOPACIO specimens (Extended Data Fig. 3b); nevertheless, even these clusters often contained fewer than 3,000 cells (Fig. 3c). Agglomerative hierarchical clustering generated six meta-clusters (Fig. 3d), but the heatmap revealed an unusual dichotomy of very bright signals for some markers and dim signals for others. Only meta-cluster C, which comprised 57% of cells in the dataset, exhibited graded signals across all channels (Fig. 3d,e). Image patches from a random set of 48 clusters revealed the presence of numerous tissue and imaging artifacts, including bright fluorescent signals, oversaturated nuclear stains and poor segmentation (Fig. 3f–h and Online Supplementary Fig. 4). Cluster 15 (meta-cluster A) arose from an image alignment error at the bottom of TOPACIO specimen 55 (Extended Data Fig. 3c) and meta-clusters B, D, E and F were caused by the presence of cells with channel intensities at or near zero as a result of image background subtraction (see Supplementary Note 1 and Supplementary Fig. 2 for a discussion of problems associated with this image processing technique).

To estimate the prevalence of visible artifacts in Dataset 1, we generated a set of downsampled single-channel images with tile gridlines superimposed to manually estimate the number of tiles impacted by overt artifacts (Online Supplementary Fig. 5). This showed that ~5,490 of the 156,300 total tiles (3.5%) were affected by antibody aggregates, tissue folds, illumination aberrations and/or slide debris. The FOXP3 channel was the worst affected (>30% of tiles; Fig. 3i) involving streaks of nonspecific antibody signal. Artifacts were less abundant in tissue resections compared to fine-needle and punch-needle biopsies (one-way analysis of variance (ANOVA), Tukey's honestly significant difference (HSD): $P_{adj} = 0.0029$–$0.0145$) but were uncorrelated with response to therapy ($F = 0.40$, $P = 0.67$; Fig. 3j). We conclude that the presence of imaging artifacts in the TOPACIO dataset causes single-cell analysis methods to fail but that errors were not preferentially biased with respect to patient response.

## Identifying and removing noisy single-cell data with CyLinter

To remove imaging artifacts from tissue images via computer-assisted human review, we developed CyLinter as a plugin for the Napari multi-channel image viewer[23] (Fig. 4a and Extended Data Fig. 4). CyLinter consists of a set of quality control (QC) software modules written in the Python programming language that process images and corresponding single-cell data in a flexible manner in which modules can be run iteratively while bookmarking progress within and between modules. CyLinter takes four files as input for each tissue specimen: (1) a stitched and registered multiplex image (TIFF/OME-TIF), (2) a cell identification mask generated by a segmentation algorithm, (3) a binary image showing the boundaries between segmented cells and (4) a spatial feature table[20] in comma-separated values (CSV) format comprising the location and computed signal intensities for each segmented cell (Fig. 4b–e, respectively). With a dataset comprising multiple images and spatial feature tables, CyLinter automatically aggregates the data into a single Pandas (Python) dataframe[31] for efficient processing

**Fig. 3 | Evaluation of pre-QC cell clustering results from Dataset 1 (TOPACIO).** **a**, UMAP embedding of ~$3 \times 10^6$ cells drawn randomly from the ~$1.9 \times 10^7$ total segmented nuclei to reduce computing time colored by HDBSCAN cluster. Black scatter points represent unclustered (ambiguous) cells. **b**, Silhouette scores for TOPACIO clusters shown in **a**. **c**, Line plot showing cell counts per TOPACIO cluster. Clusters with cell counts below the horizonal dashed red line are those with fewer than 3,000 cells highlighted in the TOPACIO embedding (inset) by red scatter points at their relative positions. **d**, Mean signal intensities of clustering cells in the pre-QC TOPACIO dataset normalized across clusters (row-wise). Six meta-clusters defined by the heatmap dendrogram at the left are highlighted. **e**, TOPACIO embedding colored by meta-clusters shown in **d**. **f**–**h**, Top three most highly expressed markers (1, green; 2, red; 3, blue) for clusters 4 (**f**), 174 (**g**) and 197 (**h**), which were all severely affected by dataset noise. A single white pixel at the center of each image highlights the reference

cell. Nuclear segmentation outlines (translucent white outlines) and Hoechst (gray) are shown for reference. **i**, The average percentage of image tiles affected by visual artifacts in each channel among the 25 TOPACIO specimens. **j**, Stacked bar chart showing the cumulative percentage of channel-specific image tiles affected by visual artifacts per TOPACIO specimen. Note that, because these data represent cumulative percentages across imaging channels, the total $y$-axis percentage may supersede 100%. Inset shows an example illumination aberration in the CD163 channel of TOPACIO specimen 73. Categories for tissue biopsy method and patient treatment response are indicated below each specimen. Artifacts were less abundant in tissue resections compared to fine-needle and punch-needle biopsies as determined by one-way ANOVA followed by pairwise Tukey's HSD ($F = 10.27$, $P = 0.0007$; fine-needle versus resection mean difference 204.83, $P_{adj} = 0.0145$; resection versus punch-needle mean difference −283.0, $P_{adj} = 0.0029$).

(Extended Data Fig. 4a). CyLinter then sensors artifactual cells from the single-cell dataframe (see https://labsyspharm.github.io/cylinter/ for implementation details).

The first CyLinter module, selectROIs (Extended Data Fig. 4b), lets the user view a multi-channel image and manually identify artifacts such as regions of tissue damage, antibody aggregates and large

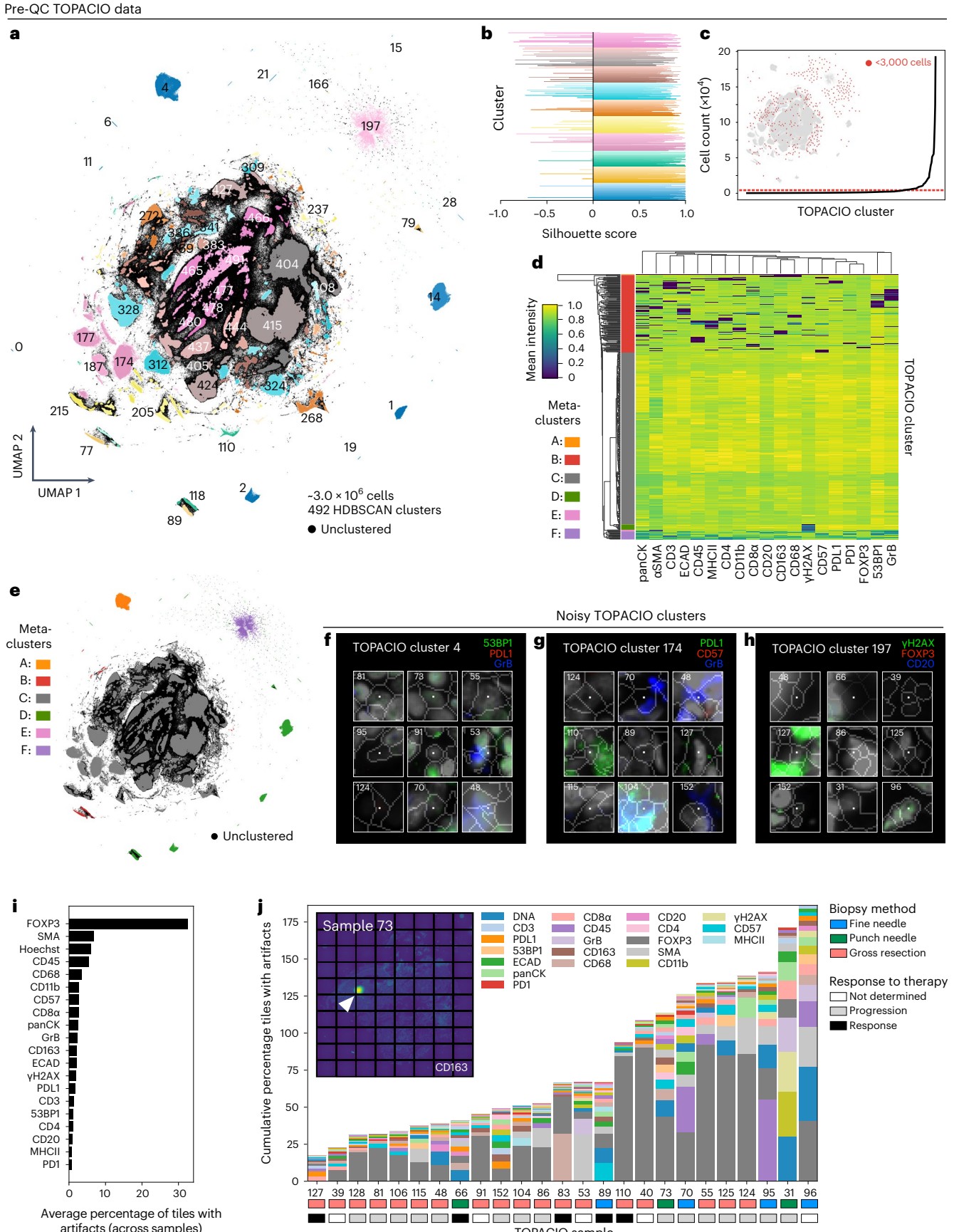

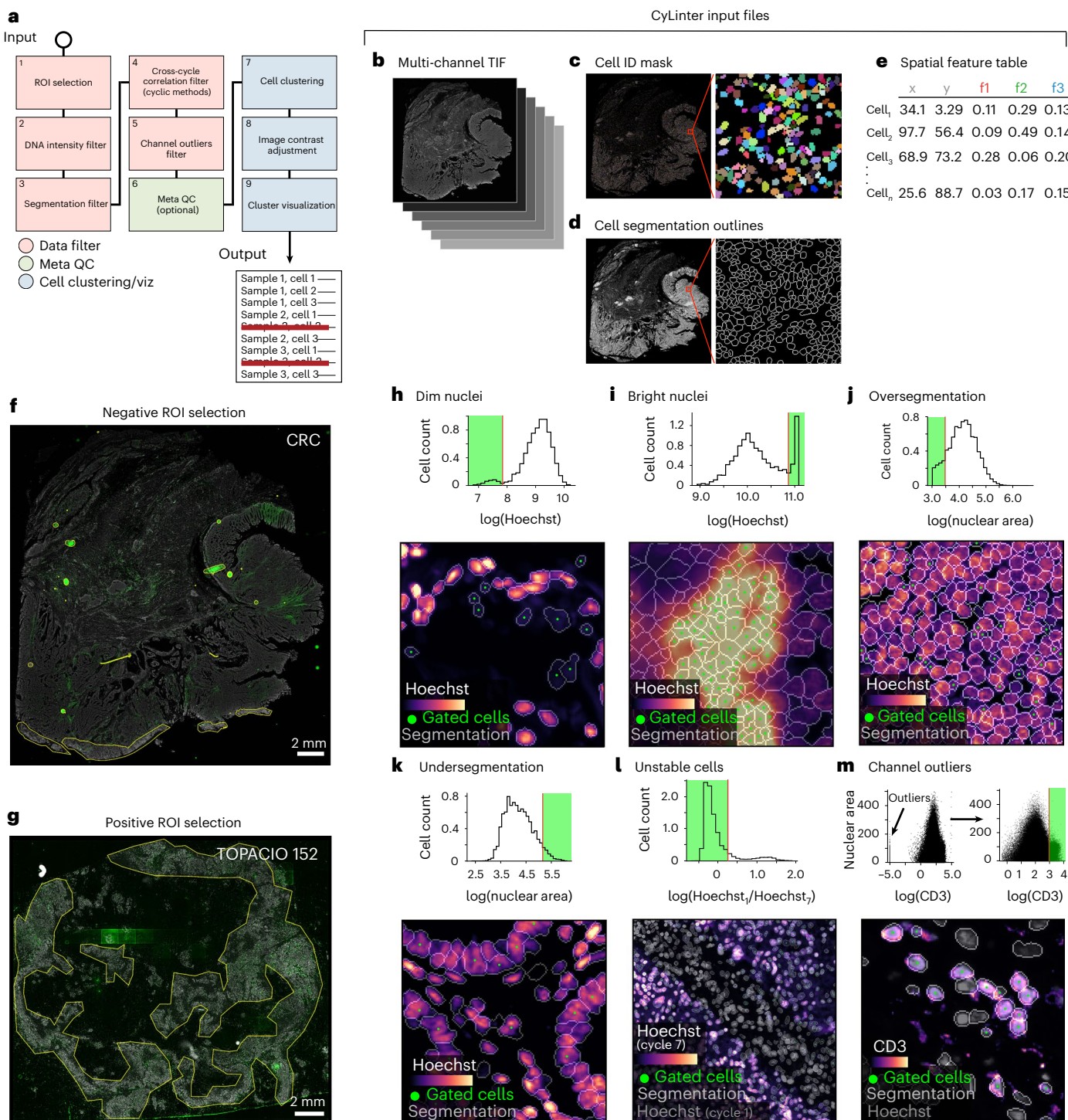

**Fig. 4 | Identifying and removing noisy single-cell data points with CyLinter.**
**a**, Schematic representation of the CyLinter workflow with modules colored by type; viz, visualization. **b**–**e**, CyLinter input files. **f**, Demonstration of negative ROI selection in CyLinter. Dataset 2 (CRC) is shown with ROIs (yellow outlines) applied to various artifacts in the CD163 channel to be dropped from subsequent analysis. **g**, Demonstration of positive ROI selection in CyLinter. Dataset 1 (TOPACIO, specimen 152) is shown with ROIs (yellow outlines) applied to regions devoid of artifacts in the FOXP3 channel to be retained for further analysis. **h**–**l**, Data filtration techniques implemented by CyLinter for the removal of dim nuclei (**h**, as demonstrated in EMIT TMA, core 12, nonneoplastic lung), bright nuclei (**i**, as demonstrated in TOPACIO, specimen 110), oversegmented nuclei (**j**, as demonstrated in the CRC image), undersegmented nuclei (**k**, as demonstrated in EMIT TMA, core 84, nonneoplastic colon) and unstable nuclei (**l**, as demonstrated in EMIT TMA, core 74, renal cell carcinoma). The top plots

show density histograms of mean Hoechst signal for cells in the given tissue. The bottom images show Hoechst signal (colormap) in a region of the same tissue with cells falling within the green region in the above histogram highlighted by green points. Nuclear segmentation outlines are shown for reference in all cases (translucent outlines). Note that, unlike **h**–**k**, which highlight cells to be excluded from analysis, cells highlighted in **l** are to be retained for further analysis. **m**, Filtering channel outliers. Top: scatter plot showing CD3 (*x* axis) versus nuclear segmentation area (*y* axis) of cells from TOPACIO specimen 152 before (left) and after (right) outlier removal. Bottom: CD3 (colormap) and Hoechst (gray) signals in a region of the same specimen with CD3⁺ cells (green points) falling to the right of the red gate in the scatter plot in which outliers have been removed. Nuclear segmentation outlines (translucent outlines) shown for reference.

illumination aberrations. Lasso tools native to the Napari image viewer are used to define regions of interest (ROIs) corresponding to artifacts. We found that negative selection (in which highlighted cells are dropped from further analysis) worked effectively for Dataset 2 (CRC; Fig. 4f), but Dataset 1 (TOPACIO) was affected by too many artifacts for this approach to be efficient. Thus, CyLinter implements an optional positive ROI selection mode, in which users select tissue regions devoid of artifacts for retention in the dataset (Fig. 4g). CyLinter also includes an automated companion algorithm that works with the selectROIs module to programmatically flag likely artifacts for human review (Extended Data Fig. 4b and Methods). This efficiently identifies features with intensities outside the distribution of biological signals.

CyLinter's dnaIntensity module (Extended Data Fig. 4c) allows users to inspect histogram distributions of per-cell mean nuclear intensities. Nuclei at the extreme left side of the distribution often correspond to truncated cells due to tissue sectioning or those lying outside of the focal plane (Fig. 4h), and those to the right side correspond to cells oversaturated with DNA dye or found in tissue folds (Fig. 4i). This module redacts data based on lower and upper thresholds initially defined by Gaussian mixture models that can be manually refined if needed. Instances of substantial over- and undersegmentation can be identified on the basis of the area of each segmentation instance followed by their removal using the dnaArea module (Extended Data Fig. 4d). This method was particularly effective at removing many oversegmented cells in the CRC image (Fig. 4j) and undersegmented cells frequently encountered among tightly packed columnar epithelial cells in normal colon specimens (for example, EMIT TMA core 84; Fig. 4k).

In cyclic imaging methods, nuclei are reimaged every cycle and individual cells are sometimes lost due to tissue movement or degradation[32,33]. CyLinter's cycleCorrelation module (Extended Data Fig. 4e) computes histograms of $\log_{10}$-transformed DNA intensity ratios between the first and last imaging cycles ($\log_{10}(DNA_1/DNA_n)$); cells that remain stable give rise to ratios around zero, whereas those that are lost give rise to a discrete peak with ratios >0. Gating the resulting histogram on stable cells eliminates unstable cells from the dataframe (Fig. 4l). Protein signals are then $\log_{10}$-transformed (Extended Data Fig. 4f). The pruneOutliers module makes it possible to visualize scatter plots of per-cell signals from all specimens in a multi-image dataset and remove residual artifacts (for example, small antibody aggregates) based on lower and upper percentile cutoffs (Fig. 4m and Extended Data Fig. 4g). Cells falling outside of the thresholds can be visualized to ensure that selected data points are indeed artifacts.

The dnaIntensity, dnaArea, cycleCorrelation and pruneOutliers modules all provide a linked view of the original image in which cells to be included or excluded by the user's chosen threshold settings are directly overlaid for visual confirmation of threshold accuracy. These labels are dynamically updated as the thresholds are adjusted. This 'visual review' is crucial to ensuring that true cell populations that happen to have extreme variations in size or signal intensity are not accidentally removed.

### Correcting for QC bias via unsupervised clustering

Human-guided artifact detection is subject to errors and biases, and the metaQC module (Extended Data Fig. 4h) addresses this by performing unsupervised clustering on equal size combinations of redacted and retained data. Cells flagged for redaction that fall within predominantly clean clusters in retained data can be added back to the dataset, while those retained in the dataset that co-cluster with predominantly noisy cells (presumed to have been missed during QC) can be removed from the dataframe. The PCA module (Extended Data Fig. 4i) performs Horn's parallel analysis to help the user determine whether two or three principal components should be used in the clustering module (described below). The setContrast module (Extended Data Fig. 4j) allows users to adjust per-channel image contrast on a reference image and apply selected settings to all images in a batch. Like the metaQC module,

CyLinter's clustering module (Extended Data Fig. 4k) allows users to perform UMAP[24] or t-distributed stochastic neighbor embedding (t-SNE)[34] data dimensionality reduction and HDBSCAN[25] density-based clustering to identify discrete cell populations in high-dimensional feature space; the clustermap module (Extended Data Fig. 4l) generates protein expression profiles for each cluster. To determine whether statistical differences exist in cell type frequency between tissues associated with test and control conditions (for example, treated versus untreated), the sampleMetadata field in CyLinter's configuration file can be populated and the frequencyStats module (Extended Data Fig. 4m) can be run. The curateThumbnails module (Extended Data Fig. 4n) automatically draws cells at random from each identified cluster and generates image galleries for efficient visual inspection. Together, these QC steps allow a user to apply a series of objective criteria to redacted and retained data to revise the output of the prior data filtration modules. On completion of the QC pipeline, CyLinter returns a single redacted spatial feature table consisting of data from multiple specimens together with a QC report for reproducibility and transparency of analysis. Artifacts identified by CyLinter are ideal for training machine learning models that can automate artifact detection; we have therefore created a public repository for the curation of CyLinter QC reports to expand our artifact library (see Supplementary Note 2 and Supplementary Fig. 3 for a description of a DL model for automated artifact detection).

### Impact of CyLinter QC on whole-slide multiplex immunofluorescence data

Applying CyLinter to Dataset 2 (CRC) resulted in the removal of ~23% of total cells (Fig. 5a). Oversegmentation was the largest problem, affecting ~16% of cells (Extended Data Fig. 5a), with ~4% or less dropped by the other QC modules. Thus, better segmentation would in principle have allowed ~93% of the data to be retained. Using HDBSCAN in CyLinter's clustering module, we identified 78 clusters (Fig. 5b)−56 more than pre-QC data (Fig. 2a). Silhouette scores were predominantly positive, suggesting effective clustering (Fig. 5c). Agglomerative hierarchical clustering yielded six meta-clusters with marker expression patterns corresponding to populations of tumor cells (meta-cluster A; Fig. 5d), stromal cells (B), memory T cells (C), macrophages (D), B cells (E) and effector T cells (F). Using the curateThumbnails module, we confirmed that all 78 clusters were largely free of visual artifacts (Fig. 5e−g and Online Supplementary Fig. 6). The increase in the number of clusters in the post-QC CRC embedding appeared to be due to the removal of pre-QC outliers that constrained the remainder of cells to a relatively narrow region of UMAP feature space. For example, by coloring the pre-QC embedding by post-QC CRC clusters, we found that pre-QC cluster 6 (Fig. 2a−d) consisted of nine different cell populations from the post-QC embedding (Fig. 5h−j). These included vimentin+ mesenchymal cells (post-QC cluster 9), memory CD8+ T cells (post-QC cluster 51) and collagen IV+ stromal cells (post-QC cluster 54). Similar analyses performed on Dataset 6 (CODEX) showed comparable improvements in the post-QC UMAP embedding, HDBSCAN clustering and associated heatmap of cluster protein expression profiles (Extended Data Fig. 5b−h and Online Supplementary Fig. 7). We conclude that post-QC clusters represent bona fide cell states that are better distributed across biologically meaningful regions of the UMAP embedding.

Despite improvements in post-QC clustering of Dataset 2 (CRC), visual inspection of the clustered heatmap (Fig. 5d) continued to reveal cells with unexpected marker expression patterns. For example, post-QC cluster 13 contained cells with epithelial markers such as keratin and ECAD and T cell markers such as CD3, CD45RO, CD45 and CD8α (Fig. 5k). There is no known cell type that expresses this marker combination. Visual inspection showed that cluster 13 actually consisted of CD8+ T cells surrounded by keratin positive tumor cells (Fig. 5l). Because segmentation is not perfect, pixels from CD8+ T cells were incorrectly assigned to neighboring epithelial cells and vice versa, a phenomenon referred to as spatial crosstalk (or lateral

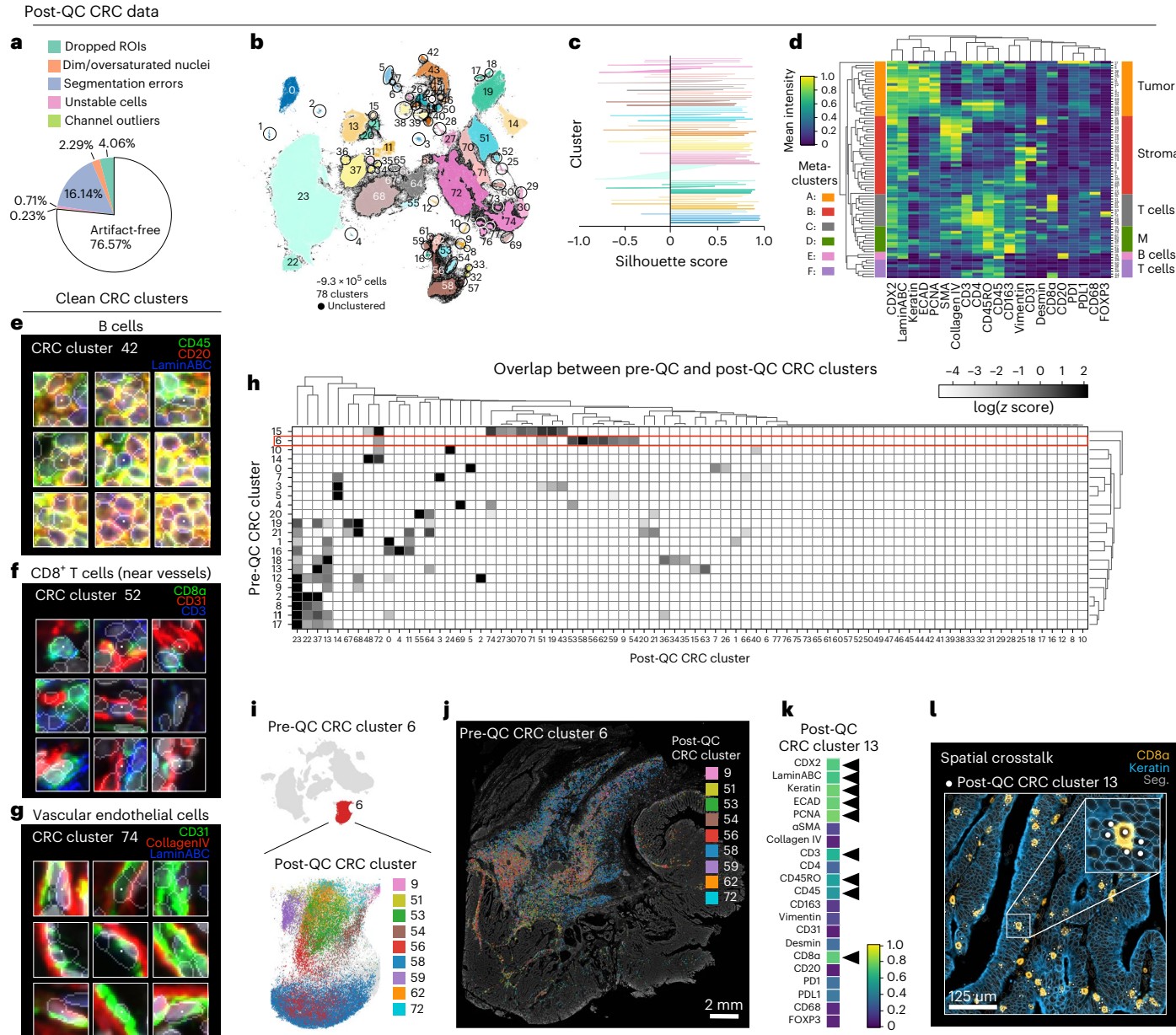

**Fig. 5 | Cleaning Dataset 2 (CRC) with CyLinter. a**, Fraction of cells in Dataset 2 redacted by each QC filter in the CyLinter pipeline. Dropped ROIs, cells dropped by selectROIs module; dim/oversaturated nuclei, cells dropped by dnaIntensity module; segmentation errors, cells dropped by areaFilter module; unstable cells, cells dropped by cycleCorrelation module; channel outliers, cells dropped by pruneOutliers module; artifact-free, cells remaining after QC. **b**, UMAP embedding of post-QC CRC data showing ~9.3 × 10⁵ cells colored by HDBSCAN cluster. Black scatter points represent unclustered (ambiguous) cells. **c**, Silhouette scores for post-QC CRC clusters shown in **b**. **d**, Mean signal intensities for clustering cells in post-QC CRC data normalized across clusters (row-wise). Six meta-clusters defined by the clustered heatmap dendrogram at the left are highlighted. **e**–**g**, Top three most highly expressed markers (1, green; 2, red; 3, blue) for post-QC CRC clusters 42 (B cells, **e**), 52 (CD8⁺ T cells near blood vessels—formed as a byproduct of spatial crosstalk, **f**) and 74 (vascular endothelial cells, **g**). A single white pixel at the center of each image highlights the

reference cell. Nuclear segmentation outlines (translucent outlines) and Hoechst (gray) shown for reference. **h**, Overlap between pre-QC CRC clusters (rows) and post-QC CRC clusters (columns) showing a one-to-many correspondence between pre- and post-QC clusters. **i**, Pre-QC CRC embedding showing the position of cluster 6 (red, inset) and its composition according to post-QC CRC clusters. **j**, Locations of cells in pre-QC cluster 6 colored by their post-QC cluster labels revealing that pre-QC cluster 6 was in fact composed of multiple cell states occupying distinct regions throughout the muscularis propria of the CRC image—a noncancerous, smooth muscle-rich region of tissue. **k**, Mean signal intensities for post-QC CRC cluster 13 cells. The black arrows highlight bright channels consistent with both proliferating epithelial cells and CD8⁺ T cells. **l**, Post-QC CRC cluster 13 cells (white points) shown in context of the CRC image demonstrating more than 30 instances of spatial crosstalk between keratin⁺ tumor cells (blue) and CD8⁺ T cells (orange). Nuclear segmentation outlines (translucent outlines) shown for reference.

spillover)[35]. Tools such as REDSEA[35] attempt to address this problem, but instances of crosstalk must currently be identified in post-QC data through inspection of heatmaps and cell image galleries.

In the case of Dataset 1 (TOPACIO), CyLinter removed 84% of cells, with most (~53%) removed during positive ROI selection (Fig. 6a). Bright outliers primarily attributed to antibody aggregates (~14% of cells),

cell detachment with increasing cycle number (12%), segmentation errors (4%) and dim/oversaturated nuclei (1%) were also common in this dataset. Cells redacted by CyLinter for both the CRC and TOPACIO datasets exhibited no discernable pattern in spatial location (Extended Data Fig. 6a,b), and data redacted from the TOPACIO specimens was not biased with respect to biopsy type (one-way ANOVA, $F = 1.93$, $P = 0.17$) or

Post-QC TOPACIO data

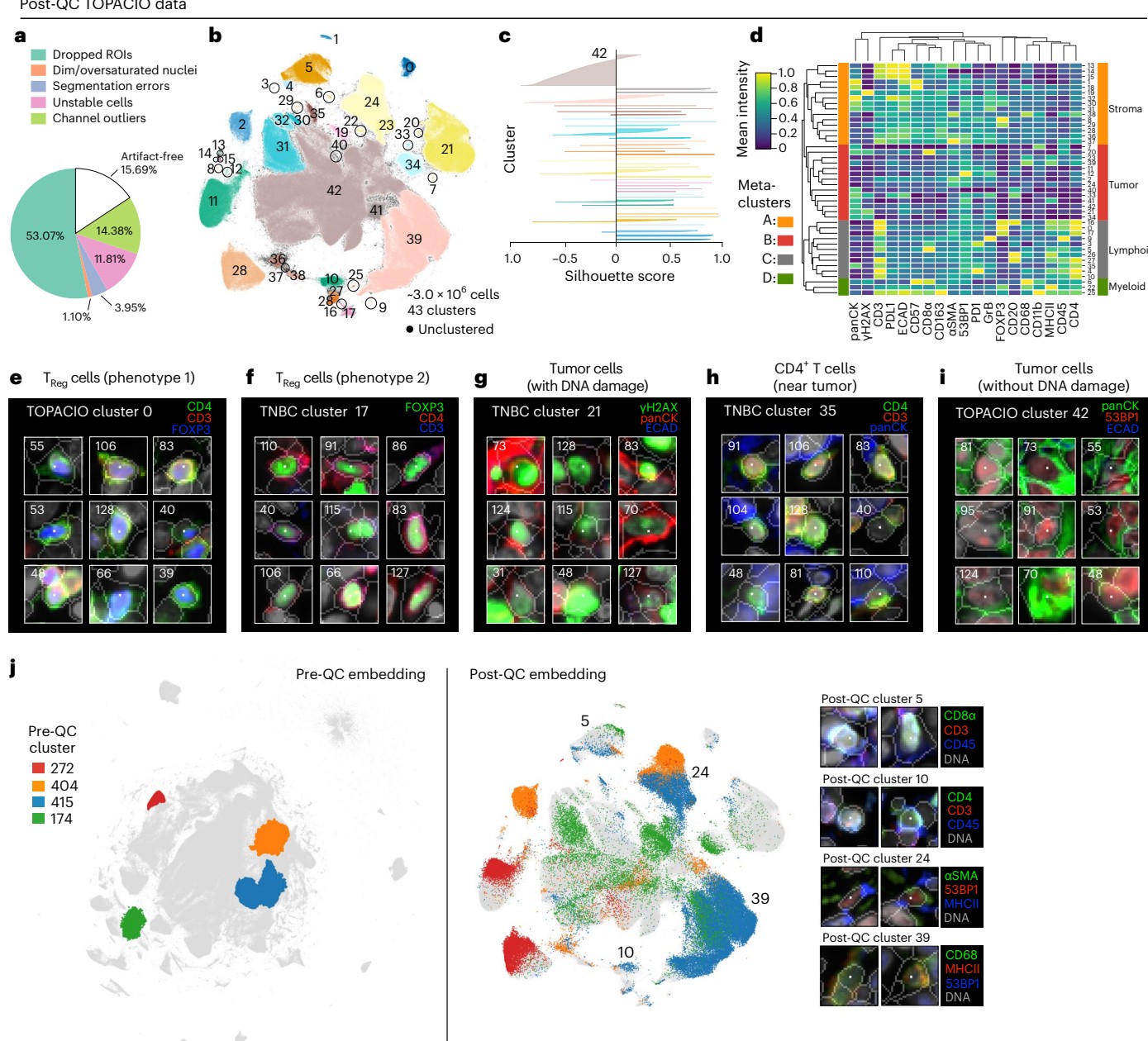

**Fig. 6 | Cleaning Dataset 1 (TOPACIO) with CyLinter. a,** Fraction of cells in the TOPACIO dataset redacted by each QC filter in the CyLinter pipeline. Dropped ROIs, cells dropped by selectROIs module; dim/oversaturated nuclei, cells dropped by dnaIntensity module; segmentation errors, cells dropped by areaFilter module; unstable cells, cells dropped by cycleCorrelation module; channel outliers, cells dropped by pruneOutliers module; artifact-free, cells remaining after QC. **b,** UMAP embedding of TOPACIO data showing ~3.0 × 10⁶ cells colored by HDBSCAN cluster. Black scatter points represent unclustered (ambiguous) cells. **c,** Silhouette scores for post-QC TOPACIO clusters shown in **b** revealing cluster 42 as an underclustered population. **d,** Mean signal intensities for clustering cells in the post-QC TOPACIO dataset normalized across clusters (row-wise). Four meta-clusters defined by the clustered heatmap dendrogram at the left are highlighted. **e–i,** Top three most highly expressed markers (1, green; 2, red; 3, blue) for clusters 0 (T_Reg cells: phenotype 1, **e**), 17 (T_Reg cells: phenotype 2, **f**), 21 (breast cancer cells with DNA damage, **g**), 35 (CD4⁺ T cells near breast cancer cells, **h**) and 42 (breast cancer cells without DNA damage, **i**). A single white pixel at the center of each image highlights the reference cell. Nuclear segmentation outlines (translucent outlines) and Hoechst (gray) shown for reference. **j,** Left: pre-QC TOPACIO UMAP embedding (also shown in Fig. 3a) with the location of five clusters selected at random highlighted. Right: location of the cells from the four pre-QC clusters shown in the embedding at left in the context of the post-QC TOPACIO UMAP embedding (also shown in **b**) demonstrating that these pre-QC clusters in fact consisted of multiple cell states. Far right: image patches of cells representing post-QC clusters 5, 10, 24 and 39.

treatment response ($F = 0.71$, $P = 0.50$). Overall, the post-QC TOPACIO dataset comprised 43 clusters among ~3.0 × 10⁶ cells (Fig. 6b). Silhouette analysis revealed positive scores for all clusters except 42, which represented the majority of tumor cells in these specimens (Fig. 6c). We found that tumor cell populations tended to cluster by patient, whereas immune cell populations tended to be more heterogenous with respect

to patient ID (Extended Data Fig. 6c). Agglomerative hierarchical clustering based on mean marker intensities yielded four meta-clusters corresponding to stromal (meta-cluster A; Fig. 6d), tumor (B), lymphoid (C) and myeloid (D) cells. CyLinter's curateThumbnails module revealed that most cells had a high degree of concordance in morphology and marker expression and were consistent with known cell types (Fig. 6e–i

and Online Supplementary Fig. 8). For example, post-QC TOPACIO cluster 0 corresponded to cells with small round nuclei with intense plasma membrane staining for CD4 and nuclear staining for FOXP3, consistent with T regulatory cells ($T_{Regs}$, Fig. 6e), cells in cluster 21 were high in panCK and γH2AX, indicative of breast cancer cells containing DNA damage (Fig. 6g), and cells in cluster 35 were conventional CD4$^+$ helper T cells ($T_{Cons}$) adjacent to panCK$^+$ tumor cells (captured as a manifestation of spatial crosstalk; Fig. 6h). Like in Dataset 2 (CRC), by coloring the post-QC embedding by pre-QC cluster labels, we found that many pre-QC clusters were composed of different post-QC cell types (Fig. 6j). For example, pre-QC cluster 415 consisted of CD8$^+$ T cells (which mapped to post-QC cluster 5), CD4$^+$ T cells (post-QC cluster 10), αSMA$^+$ stromal cells (post-QC cluster 24) and CD68$^+$ macrophages (post-QC cluster 39). Thus, imaging artifacts in the TOPACIO data not only resulted in an unrealistically large number of clusters, but these clusters still contained mixed cell types.

## Discussion

In this paper, we show that artifacts commonly present in highly multiplexed images of tissue have a dramatic impact on single-cell analysis. These artifacts can be broadly subdivided into three categories: (1) those intrinsic to the specimen itself such as tissue folds and hair or lint, (2) those arising during staining and image acquisition such as antibody aggregates and (3) those arising during image-processing such as cell segmentation errors. The first class is unavoidable and does not usually interfere with visual review by human experts. The second and third classes can be minimized but not fully eliminated by good experimental practices. However, even relatively few artifacts can strongly impact clustering and other types of single-cell analysis as demonstrated by Datasets 2 (CyCIF) and 6 (CODEX) in this study. Archival specimens stored in paraffin blocks or mounted on slides years before imaging are even more problematic insofar as artifacts are common and only one slide may be available for each specimen; unfortunately, this is not unusual in correlative studies of completed clinical trials.

The presence of cells affected by imaging artifacts has complex effects on clustering algorithms used to identify cell types and states. Artifacts can lead to the generation of large numbers of spurious clusters and also cause clusters to contain cells of multiple types. Removing the problematic cells using CyLinter solves this problem. When data are removed, there is always concern that findings will be biased. CyLinter addresses this in several ways by allowing for visual review of filtered cells against the image itself, performing meta-analysis of redacted features (metaQC), performing specimen subgroup analysis and generating a comprehensive QC report for each set of specimens analyzed; the latter should ideally be included with all publicly deposited datasets. Similar issues arise with single-cell sequencing, although much of the problem occurs during tissue dissociation, microfluidic or flow cytometry sorting, and library preparation[36,37]. An advantage of tissue imaging is that redacted data can be inspected in the context of the original image to identify patterns indicative of selection bias.

QC is recognized as a critical step in the acquisition of scRNA-seq data, and a robust ecosystem of QC tools has therefore been developed[36,38]. In contrast, CyLinter is among the first tools for QC of highly multiplexed tissue images. CyLinter is designed to accelerate and systematize human visual review, making it compatible with a wide range of tissue types. Efficiency is increased through automated ROI curation, smart thresholding using Gaussian mixture models and use of multi-specimen dataframes. We found that even the severely affected set of 25 tissue specimens represented by the TOPACIO dataset took a single reviewer less than a week to clean, which compares favorably with several weeks needed to collect the data, and several months or more to perform detailed spatial analysis. However, the necessity for human review is a potential weakness when working with very large datasets. As a first step in automating image QC, we describe a proof-of-concept DL model for automated artifact detection (Supplementary Note 2).

The area under the receiver operator curve of ~0.73 shows that the approach is feasible but that performance is not yet adequate for general use, probably due to insufficiently diverse training data. CyLinter is the ideal way to generate this training data, and we have therefore created a public artifact repository linked to the CyLinter website. This repository should also address limitations in the scope of the artifacts we analyze, particularly as new imaging methods are developed. In its current form, CyLinter is a stand-alone software package used on workstations with limited power and memory. To overcome this limitation, future versions will run on scalable cloud infrastructures that also facilitate collaboration and are easier to update.

Microscopy is traditionally a visual field, and our experience with over 1,000 whole-slide high-plex images from dozens of tissue and tumor types has demonstrated that spatial feature tables generated using existing algorithms not only contain errors and omissions, but they also poorly represent much of the morphological information in images. This emphasizes the necessity of visual review: any hypothesis generated through analysis of data in a spatial feature table must be confirmed through inspection of the underlying images. At the same time, visual review must be backed up by objective methods that detect and correct for human errors and biases. The QC tools in CyLinter achieve this combination of human review and algorithmic backup and represent one key step in making single-cell analysis of high-plex spatial profiles more interpretable and reproducible.

## Online content

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

## Methods

### CyLinter software

CyLinter software is written in Python3, version controlled on Git/GitHub (https://github.com/labsyspharm/cylinter)[39], validated for Mac, PC, and Linux operating systems and archived on the Anaconda package repository. The tool can be installed at the command line using the Anaconda package installer (see the CyLinter website https://labsyspharm.github.io/cylinter/ for details) and is executed with the following command: *cylinter cylinter_config.yml*, where *cylinter_config.yml* is an experiment-specific YAML configuration file. An optional --*module* flag can be passed before specifying the path to the configuration file to begin the pipeline at a specified module. More details on configuration settings can be found at the CyLinter website and GitHub repository. The tool uses the Napari image viewer for image browsing and annotation tasks. The tool also uses numerical and image-processing routines from multiple Python data science libraries, including pandas, numpy, matplotlib, seaborn, SciPy, scikit-learn and scikit-image. OME-TIFF files are read using tifffile and processed into multi-resolution pyramids using a combination of Zarr and dask routines that allow for rapid panning, zooming and processing of large (hundreds of gigabytes) images. The CyLinter pipeline consists of multiple QC modules, each implemented as a Python function, that perform different visualization, data filtration or analysis tasks. Several modules return redacted versions of the input spatial feature table, while others perform analysis tasks such as cell clustering. CyLinter is freely available for academic reuse under the MIT license. A minimal example dataset consisting of four tissue cores from the EMIT TMA (Dataset 3) used in this study can be downloaded from the Synapse data repository (Synapse ID: syn52859560) by following instructions at the CyLinter website (https://labsyspharm.github.io/cylinter/exemplar/). All CyLinter analyses presented in this work were performed on a commercially available 2019 MacBook Pro equipped with eight 2.4 GHz Intel Core i9 processors (5.0 GHz Turbo Boost) and 32 GB 2,400 MHz DDR4 memory. Imaging data analyzed in this study were stored on and accessed from an external hard drive with 12TB capacity. Implemented software versions were as follows: CyLinter v0.0.46–v0.0.49, Python v3.8–v3.11.

### t-CyCIF

The CyCIF approach to multiplex imaging involves iterative cycles of antibody incubation with tissue, imaging and fluorophore deactivation as described previously[2]; protocols and methods related to CyCIF are available on Protocols.io (see 'Detailed experimental protocols' section). Briefly, multiplex CyCIF images were collected using a RareCyte CyteFinder II HT Instrument equipped with a 20× (0.75 numerical aperture (NA)) objective and 2 × 2 pixel binning. This setup allowed for the acquisition of four-channel image tiles with dimensions 1,280 × 1,080 pixels and a corresponding pixel size of 0.65 μm per pixel. All four channels are imaged during each round of CyCIF, one of which is always reserved for nuclear counterstain (Hoechst or 4′,6-diamidino-2-phenylindole (DAPI)) to visualize cell nuclei. RCPNL files containing 16-bit imaging data were generated (one per image tile) during each imaging cycle.

### Image processing

Raw microscopy image tiles (RCPNL files) for the datasets described in this study were processed into stitched, registered and segmented OME-TIFF[40] files using the MCMICRO image-processing software[19]. Corresponding cell × feature CSV files (that is, spatial feature tables) were also generated by MCMICRO. Specific algorithms implemented in MCMICRO for the processing of each dataset are as follows: BaSiC[41] (v1.0.1)—a Fiji/ImageJ plugin for background and shading correction used to perform flatfield and darkfield image correction; ASHLAR[32] (v1.11.1)—a program for seamless mosaic image processing across imaging cycles; Coreograph (v2.2.0)—a program for dearraying TMA images into individual TIFF and CSV files per tissue core (https://github.com/

HMS-IDAC/UNetCoreograph); UnMICST[33] (v2.4.7)—an implementation of semantic cell segmentation based on the U-Net architecture[42]; S3segmenter (v1.2.0)—a watershed algorithm used in conjunction with UnMICST (https://github.com/HMS-IDAC/S3segmenter); and MCQuant (v1.3.1)—an algorithm used for per cell feature extraction including *X*,*Y* spatial coordinates, segmentation areas, mean marker intensities and nuclear morphology attributes (https://github.com/labsyspharm/quantification).

### Automated artifact detection in CyLinter with classical algorithms

An algorithm consisting of classical image analysis steps was designed to automatically identify prevalent artifacts commonly found in highly multiplexed images (for example, illumination aberrations, antibody aggregates and tissue folding). The model is applied on a channel-by-channel basis and works on downsampled versions of each channel, rescaling pixel values to uint8 bit depth for efficient processing. A series of operations in mathematical morphology consisting of erosion and local mean smoothing followed by dilation are applied to transform each downsampled image channel. These three steps utilize a disk kernel, where the kernel size is a user-defined parameter assumed to have a diameter on the order of three to five single cells, conditional on image pixel size. This kernel is then expanded to find local maxima seed points corresponding to putative artifacts. Each artifact is extracted via a flood fill operation according to a specific tolerance parameter that is adjusted in real time by the user. The union of the flood fill regions produces a binary artifact mask that is resized to the original image dimensions; cells falling within mask boundaries are then dropped from the corresponding spatial feature table.

### Deep learning-based automated artifact detection

The machine learning artifact detection model implemented in this study derives from the Feature Pyramid Network (FPN)[43], a fully convolutional encoder–decoder architecture designed for object detection tasks applicable to semantic image segmentation. The encoder network is implemented using a ResNet34 backbone[44] with model parameters initialized from the pretraining weights on ImageNet. Input image tiles of size 2,048 × 2,048 pixels (acquired at a nominal resolution of 0.65 μm per pixel) were downsampled to 256 × 256 pixels and fed into the encoder network to produce low-resolution feature maps. Resulting feature maps were then decoded into feature pyramids through iterated upsampling using a bilinear interpolation and combined with the original feature maps. Each layer of the feature pyramid was upsampled to the same resolution and segmented such that all resulting predicted artifact masks were combined to yield the final composite prediction mask. The FPN architecture is implemented using the Segmentation Models library for image segmentation based on the Python and PyTorch frameworks[45]. The model was trained using the Adam optimizer with a Dice similarity coefficient loss function and a fixed learning rate ($1 \times 10^{-4}$) using a batch size of 16 image tiles for 10 epochs.

### Dataset 1 (TOPACIO, CyCIF)

The TOPACIO dataset used in this study consists of 25 deidentified FFPE tissue sections (5 μm thick) of TNBC from patients enrolled in the TOPACIO clinical trial (ClinicalTrials.gov Identifier: NCT02657889). Specimens were collected via one of three different biopsy methods: fine needle, punch needle or gross tumor resection and procured from Tesaro and Merck & Co. as part of the recently completed trial. Slides were mounted onto Superfrost Plus glass microscope slides (Fisher Scientific, 12-550-15) then dewaxed and antigen-retrieved using a Leica BOND RX Fully Automated Research Stainer before multiplex data acquisition by CyCIF. The TOPACIO dataset was collected during this study using a CyteFinder slide scanning fluorescence microscope and its built-in image acquisition software (RareCyte). Images were

acquired at 20× magnification with 2 × 2 binning (0.65 µm per pixel nominal resolution) over 10 CyCIF cycles using 27 markers (19 plus Hoechst evaluated in this study); see Supplementary Table 1 for further details. The following antibodies were used in the acquisition of this dataset (name, clone, vendor, catalog number, RRID, dilution):

Donkey anti-Rat A488 (secondary only), polyclonal, Invitrogen, A21208, AB_2535794, 1:1,000

Donkey anti-Rabbit A555 (secondary only), polyclonal, Invitrogen, A31572, AB_162543, 1:1,000

Donkey anti-Goat A647 (secondary only), polyclonal, Invitrogen, A21447, AB_2535864, 1:1,000

CD3 (secondary conjugated), CD3-12, Abcam, ab11089, AB_2889189, 1:200

PDL1 (secondary conjugated), E1L3N, Cell Signaling Technology, 13684S, AB_2687655, 1:200

53BP1 (secondary conjugated), polyclonal, Bethyl Laboratories, A303-906A, AB_2620256, 1:200

E-Cadherin(A488), 24E10, Cell Signaling Technology, 3199S, AB_2291471, 1:400

panCK(e570), AE1/AE3, EBioscience, 41-9003-82, AB_11218704, 1:800

PD1(A647), EPR4877(2), abcam, ab201825, AB_2728811, 1:200

CD8a(A488), AMC908, EBioscience, 53-0008-82, AB_2574413, 1:200

CD45(PE), 2D1, R&D, FAB1430P, AB_2237898, 1:100

GrB(A647), 2C5, Santa Cruz, sc-8022AF647, AB_2232723, 1:200

CD163(A488), EPR14643-36, Abcam, ab218293, AB_2889155, 1:400

CD68(PE), D4B9C, Cell Signaling Technology, 79594S, AB_2799935, 1:200

CD20(e660), L26, EBioscience, 50-0202-80, AB_11151691, 1:400

CD4(A488), polyclonal, R&D Systems, FAB8165G, AB_2728839, 1:200

FOXP3(e570), 236A/E7, EBioscience, 41-4777-82, AB_2573609, 1:100

SMA(e660), 1A4, EBioscience, 50-9760-82, AB_2574362, 1:800

CD11b(A488), C67F154, EBioscience, 53-0196-82, AB_2637196, 1:150

pSTAT1(A555), 58D6, Cell Signaling Technology, 8183S, AB_10860600, 1:200

yH2AX(A647), 2F3, BioLegend, 613407, AB_2295046, 1:200

CD57(FITC), NK-1, BD, 561906, AB_395986, 1:100

Ki67(e570), 20Raj1, EBioscience, 41-5699-82, AB_11220278, 1:100

MHCII/HLA-DPB1(A647), EPR11226, Abcam, ab201347, AB_2861375, 1:400

STING(A488), EPR13130, Abcam, ab198950, AB_2889208, 1:400

pTBK1(A555), D52C2, Cell Signaling Technology, 13498S, AB_2943237, 1:200

pSTAT3(A647), D3A7, Cell Signaling Technology, 4324S, AB_10694637, 1:200

PCNA(A488), PC10, Cell Signaling Technology, 8580S, AB_2617115, 1:400

HLA-A(A555), EP1395Y, Abcam, ab207872, AB_2889202, 1:400

cPARP(A647), D64E10, Cell Signaling Technology, 6987S, AB_10699459, 1:100

## Dataset 2 (CRC, CyCIF)

The CRC dataset was previously published[15] and consists of a whole-slide section (1.6 cm²) of human colorectal adenocarcinoma tissue (section# 097) from a 69-year-old white male imaged at 20× magnification with 2 × 2 binning (0.65 µm per pixel nominal resolution) over 10 CyCIF cycles using 24 markers across 10 CyCIF cycles (21 plus Hoechst evaluated in the current study) collected as part of the Human Tumor Atlas Network (HTAN) and is available through the HTAN Data Portal (https://data.humantumoratlas.org). See Supplementary Table 1 for further details and associated identifiers.

## Dataset 3 (EMIT TMA22, CyCIF)

The EMIT TMA dataset was previously published[19] and consists of human tissue specimens from 42 patients organized as a multi-tissue array (HTMA427) under an excess tissue protocol (clinical discards) approved by the institutional review board (IRB) at Brigham and Women's Hospital (BWH IRB 2018P001627). Two 1.5-mm-diameter cores were acquired from each of 60 tissue regions with the goal of acquiring one or two examples of as many tumors as possible (with matched normal tissue from the same resection when feasible). Overall, the TMA contains 123 cores including 3 'marker cores' consisting of normal kidney cortex that were added to the TMA in an arrangement that makes it possible to orient the overall TMA image. Not including the marker cores, 44 cores were from males and 76 were from females between 21 and 86 years of age. The EMIT TMA22 dataset was acquired at 20× magnification with 2 × 2 binning (0.65 µm per pixel nominal resolution) over 10 CyCIF cycles using 27 markers (20 plus Hoechst evaluated in the current study) and is available for download from the Synapse data repository (https://www.synapse.org/#!Synapse:syn22345750); see Supplementary Table 1 for further details.

## Dataset 4 (HNSCC, CODEX)

The HNSCC CODEX dataset consists of two sections of the same deidentified specimen of head and neck squamous carcinoma (HNSCC) imaged at 20× magnification with 2 × 2 binning (0.65 µm per pixel nominal resolution) over 9 imaging cycles using 15 markers plus DAPI. These data were collected by the laboratory of Kai Wucherpfennig at Dana-Farber Cancer Institute; see Supplementary Table 1 for further details.

## Dataset 5 (normal tonsil, mIHC)

The mIHC dataset was previously published[19] and consists of a deidentified whole-slide tonsil specimen from a 4-year-old female of European ancestry procured from the Cooperative Human Tissue Network (CHTN), Western Division, as part of the HTAN SARDANA Trans-Network Project and imaged at 20× magnification with 2 × 2 binning (0.5 µm per pixel nominal resolution) over 5 mIHC cycles using 18 markers plus Hoechst; see Supplementary Table 1 for further details.

## Dataset 6 (normal large intestine, CODEX, specimen 1)

A single section of deidentified human tissue from a 78-year-old African American male imaged at 20× magnification (0.75 NA, 0.38 µm per pixel nominal resolution) over 23 imaging cycles using 59 markers (58 evaluated in this study, as DRAQ5 was excluded due to its overlap with Hoechst). These data were collected at Stanford University as part of the Human BioMolecular Atlas Program (HuBMAP); see Supplementary Table 1 for further details.

## Dataset 7 (normal large intestine, CODEX, specimen 2)

The large intestine CODEX dataset consists of a single section of deidentified human tissue from a 24-year-old white male imaged at 20× magnification (0.75 NA, 0.38 µm per pixel nominal resolution) over 24 imaging cycles using 54 markers (53 evaluated in this study, as DRAQ5 was excluded due to its overlap with Hoechst). These data were collected at Stanford University as part of the Human BioMolecular Atlas Program (HuBMAP); see Supplementary Table 1 for further details.

## Detailed experimental protocols

1. FFPE Tissue Pre-treatment on Leica Bond RX V.2 (https://doi.org/10.17504/protocols.io.bji2kkge)
2. Tissue Cyclic Immunofluorescence (t-CyCIF) V.2 (https://doi.org/10.17504/protocols.io.bjiukkew)

## Ethics and IRB statement

The TOPACIO clinical trial (ClinicalTrials.gov Identifier: NCT02657889) was conducted in accordance with the ethical principles founded

in the Declaration of Helsinki and received central approval by the Dana-Farber IRB and/or relevant competent authorities at each treatment site. All patients provided written informed consent to participate in the study. Tissue specimens and metadata were deidentified for the work performed at Harvard Medical School, which complied with all relevant ethical regulations and was reviewed and approved under IRB protocol 19-0186. The research described in this study is considered non-human subjects research.

## Reporting summary

Further information on research design is available in the Nature Portfolio Reporting Summary linked to this article.

## Data availability

One primary dataset and six referenced datasets are analyzed in this study. The primary dataset encompasses 25 specimens of TNBC from patients enrolled in the TOPACIO clinical trial imaged by CyCIF (ClinicalTrials.gov ID: NCT02657889). Access to the TOPACIO dataset can be made through the explicit permission of the TOPACIO clinical trial sponsor (Tesaro, Inc.). All other datasets necessary to reproduce the findings in this study can be found at the Sage Bionetworks Synapse data repository at https://www.synapse.org/#!Synapse:syn54523217. The CRC CyCIF dataset is also available at the HTAN Data Portal (https://data.humantumoratlas.org/). See Methods for a description of each dataset and Supplementary Table 1 for complete details on the sources of each referenced dataset and their associated identifiers.

## Code availability

Source code for the CyLinter program is freely available for academic reuse under the MIT open-source license agreement at GitHub (https://github.com/labsyspharm/cylinter)[39] and continually integrates the latest versions of the following open-source data science libraries: napari, magicgui, pyqt, qtpy, pyyaml, tifffile, scikit-image, zarr, pandas, pyarrow, numpy, matplotlib, seaborn, hdbscan, umap-learn, joblib, scikit-learn, scipy, cellcutter, natsort, numba, svglib and pypdf2. The scripts used to generate the figure panels in this article are available via a dedicated GitHub repository (https://github.com/labsyspharm/cylinter-paper) archived on Zenodo (https://doi.org/10.5281/zenodo.10067803)[46].

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

## Acknowledgements

This work was supported by Ludwig Cancer Research (P.K.S. and S.S.) and by NIH NCI grants U2C-CA233280, U2C-CA233262 and U01-CA284207 (P.K.S and S.S.). Development of computational methods and image processing software is supported by the Gates Foundation grant INV-027106 (P.K.S.), a Team Science Grant from the Gray Foundation (P.K.S. and S.S.), the David Liposarcoma Research Initiative at Dana-Farber Cancer Institute supported by KBF Canada via the Rossy Foundation Fund (P.K.S. and S.S.) and the Emerson Collective (P.K.S.). S.S. is supported by the BWH President's Scholars Award. We acknowledge J. Tefft for superb editorial support; C. Yapp for help with artifact annotation; K. Wucherpfennig and S. Marx for providing the HNSCC CODEX dataset; Z. Maliga and C. Jacobson for providing CyCIF EMIT TMA22 images; and the Dana-Farber/Harvard Cancer Center for use of the Specialized Histopathology Core, which provided TMA construction and sectioning services. We also thank Y.-A. Chen for assisting in the collection of CyCIF data from the SARDANA-097 tissue specimen performed as part of the NCI HTAN. Results shown in this study are in part based upon data generated by the Human BioMolecular Atlas Program (HuBMAP, https://hubmapconsortium.org).

## Author contributions

G.J.B. conceived and designed the study. P.K.S. supervised the work and secured funding. G.J.B. developed the CyLinter software program, J.L.G. and E.A.M. provided access to the TOPACIO clinical biopsies, J.-R.L. acquired t-CyCIF data from the TOPACIO specimens, T.V. and J.A.D. curated tissue ROIs for the TOPACIO specimens, E.N. and Z.Z. developed the method for automated artifact detection, G.J.B. performed CyLinter analysis on all datasets and generated the figures, and G.J.B. and P.K.S. wrote the manuscript with input from all authors.

## Competing interests

P.K.S. is a cofounder and member of the Board of Directors of Glencoe Software, a member of the Board of Directors for Applied Biomath and a member of the Scientific Advisory Board for RareCyte, NanoString and Montai Health; he holds equity in Glencoe and RareCyte. P.K.S. is a consultant for Merck and declares that none of these relationships has influenced the content of this manuscript. E.A.M. reports compensated service on Scientific Advisory Boards for AstraZeneca, BioNTech and Merck; uncompensated service on Steering Committees for Bristol Myers Squibb and Roche/Genentech; speakers' honoraria and travel support from Merck Sharp & Dohme; and institutional research support from Roche/Genentech (via an SU2C grant) and Gilead. She also reports research funding from Susan Komen for the Cure for which she serves as a Scientific Advisor, and uncompensated participation as a member of the American Society of Clinical Oncology Board of Directors. J.L.G. serves or has previously served on advisory boards and/or as a scientific advisory board member for Array BioPharma/Pfizer, AstraZeneca, BD Biosciences, Carisma, Codagenix, Duke Street Bio, GlaxoSmithKline, Kowa, Kymera, OncoOne and Verseau Therapeutics and has research grants from Array BioPharma/Pfizer, Duke Street Bio, Eli Lilly, GlaxoSmithKline and Merck. The other authors declare no competing interests.

## Additional information

**Extended data** is available for this paper at https://doi.org/10.1038/s41592-024-02328-0.

**Correspondence and requests for materials** should be addressed to Gregory J. Baker or Peter K. Sorger.

**Peer review information** *Nature Methods* thanks Andrea Radtke and the other, anonymous, reviewer(s) for their contribution to the peer review

of this work. Primary Handling Editor: Rita Strack, in collaboration with the *Nature Methods* team. Peer reviewer reports are available.

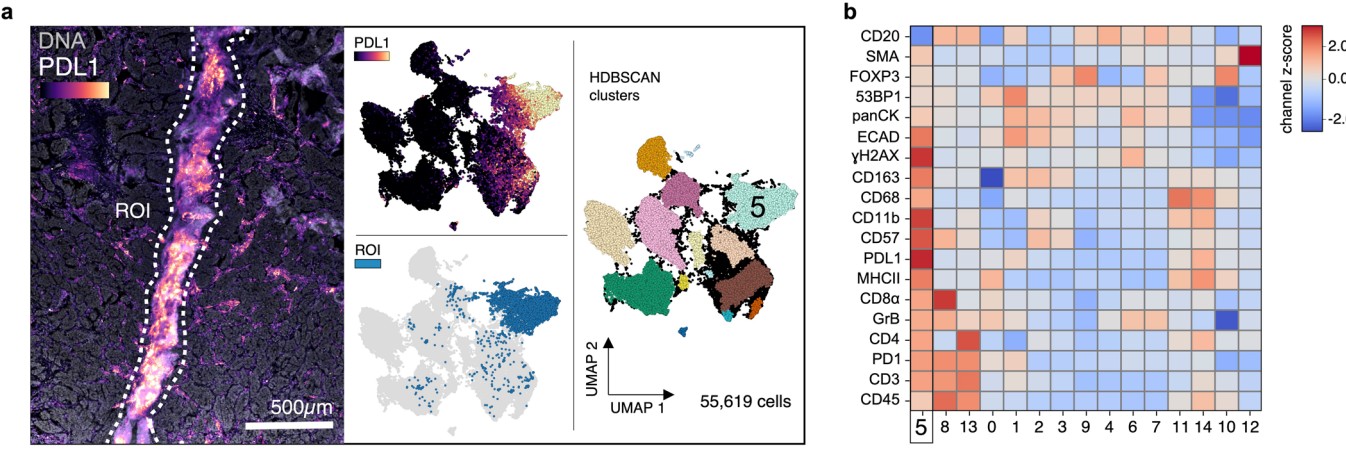

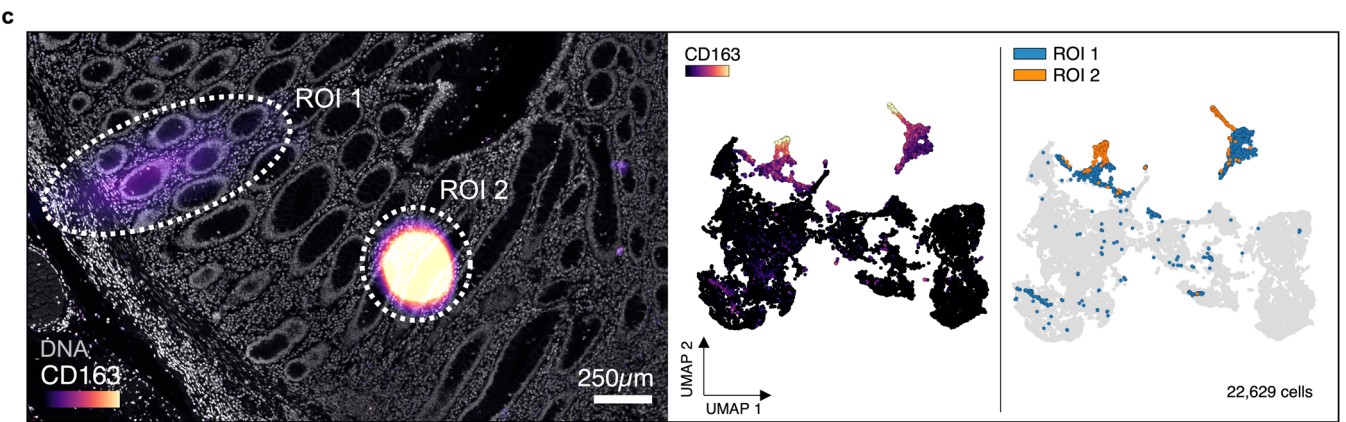

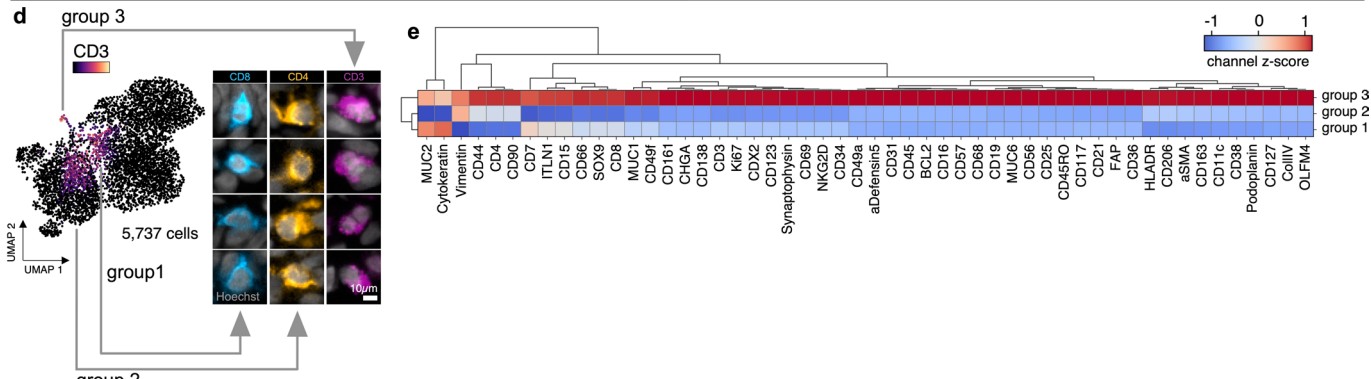

**Extended Data Fig. 1 | Recurring artifacts in whole-slide immunofluorescence images of tissue and their effects on tissue-derived, single-cell data. a**, Left: Field of view from Dataset 1 (TOPACIO, specimen 110) showing a tissue fold (ROI, dashed white outline) as viewed in channels PDL1 (colormap) and Hoechst (gray). Right: UMAP embedding of 19-channel single-cell data from the left image colored by PDL1 intensity (top left), ROI inclusion (bottom left), and HDBSCAN cluster (center right). Cells in cluster 5 are those affected by the tissue fold. **b**, Channel *z* scores for HDBSCAN clusters in from panel (a) demonstrating that cluster 5 cells (those affected by the tissue fold) are artificially bright for all channels likely due to a combination of tissue overlap and insufficient antibody washing. **c**, Left: Field of view from Dataset 2 (CRC) showing two illumination aberrations (ROIs, dashed white outlines) as viewed in channels CD163 (colormap) and Hoechst (gray). Right: UMAP embedding of 21-channel single-cell data from the left image colored by CD163 intensity (left) and inclusion in one of the two ROIs (right). **d**, UMAP embedding of the 52-channel single-cell data shown in Fig. 1j (Dataset 7, large intestine, CODEX) after cells affected by the five illumination aberrations have been removed. Three groups of cells bright for CD3 remain (groups 1–3). Image galleries at right show four examples of each cell type in representative channels: group 1 = CD8⁺ T cells, group 2 = CD4⁺ T cells, group 3 = undefined cells immunoreactive to all 52 channels. **e**, Channel *z* scores for HDBSCAN clusters in (d) demonstrating that group 3 cells are bright for all 52 channels despite not being affected by microscopy artifacts.

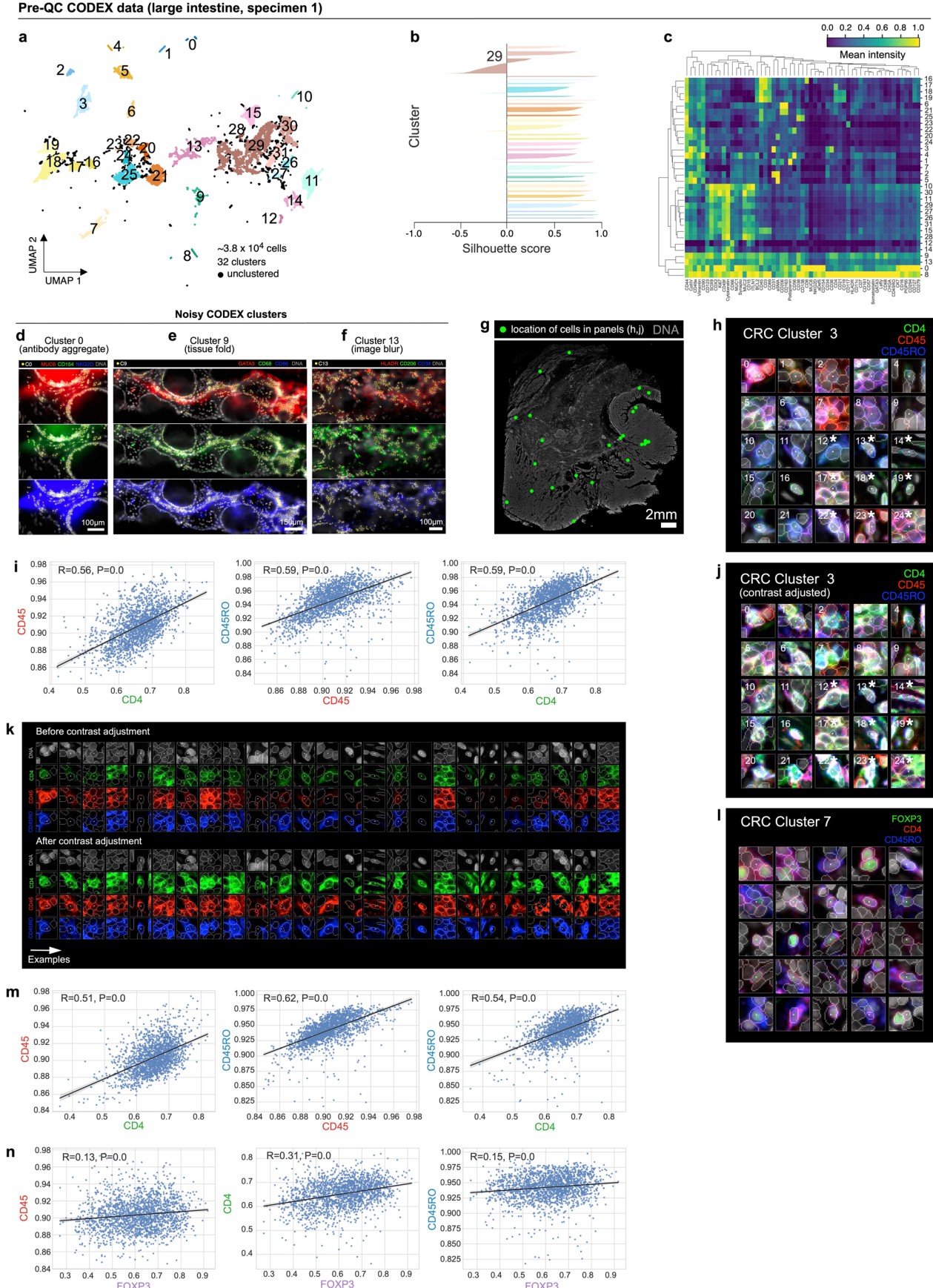

Pre-QC CODEX data (large intestine, specimen 1)

Noisy CODEX clusters

**Extended Data Fig. 2 | See next page for caption.**

**Extended Data Fig. 2 | Evaluation of pre-QC cell clustering results from Datasets 6 (large intestine, CODEX) and 2 (CRC, CyCIF). a**, UMAP embedding of Dataset 6 showing ~3.8 × 10⁴ cells colored by HDBSCAN cluster. Black scatter points represent ambiguous cells (10.5% of total). **b**, Silhouette scores for CODEX clusters in (a) revealing cluster 29 as an under-clustered population. **c**, Mean signal intensities of clustering cells from Dataset 6 normalized across clusters (row-wise). **d**, Correlated, non-specific signals in a region of Dataset 6 as seen in channels MUC6 (red), CD154 (green), and NKG2D (blue). Yellow points highlight cluster 0 cells formed due to this artifact. **e**, Tissue fold in a region of Dataset 6 as seen in channels GATA3 (red), CD68 (green), and CD66 (blue). Yellow points highlight cluster 9 cells formed due to this artifact. **f**, Image blur in a region of Dataset 6 as seen in channels HLADR (red), CD206 (green), and CD38 (blue). Yellow points highlight cluster 13 cells formed due to this artifact. **g**, Location of CRC cluster 3 cells in (g) revealing no spatial bias in the distribution of these cells. **h**, Top three most highly expressed markers (1: green, 2: red, 3: blue) for the 25 members of CRC cluster 3 (memory helper T) cells represented by the rugplots of Fig. 2n. White asterisks highlight cells shown in enlarged format in Fig. 2m.

A single white pixel at the center of each image patch highlights the reference cell. Nuclear segmentation outlines (translucent white outlines) and Hoechst (gray) shown for reference. **i**, Regression plots showing correlation among CD4, CD45, and CD45RO marker expression by 1.9 × 10³ CRC cluster 3 cells (two-sided, Pearson R, p < 0.05). **j**, CRC cluster 3 cells in (h) after signal intensity cutoffs have been adjusted per image to improve their homogeneity of appearance. White asterisks highlight cells shown in enlarged format in (Fig. 2o). **k**, CRC cluster 3 cells with channels shown separately for clarity. Top panels show cells before contrast adjustment (h), bottom panels show cells after contrast adjustment (j). **l**, Top three most highly expressed markers (1: green, 2: red, 3: blue) for 25 CRC cluster 7 (T$_{Reg}$) cells. A single white pixel at the center of each image patch highlights the reference cell. Nuclear segmentation outlines (translucent white outlines) and Hoechst (gray) shown for reference. **m**, Regression plots showing strong correlation among CD4, CD45, and CD45RO marker expression of 1.9 × 10³ CRC cluster 7 cells (two-sided, Pearson R, p < 0.05). **n**, Regression plots showing weak correlation between FOXP3 and CD4, CD45, and CD45RO marker expression of 1.9 × 10³ CRC cluster 7 cells (two-sided, Pearson R, p < 0.05).

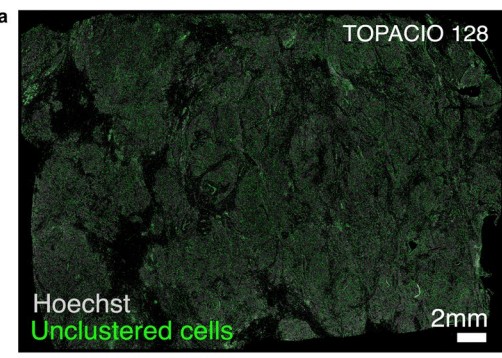

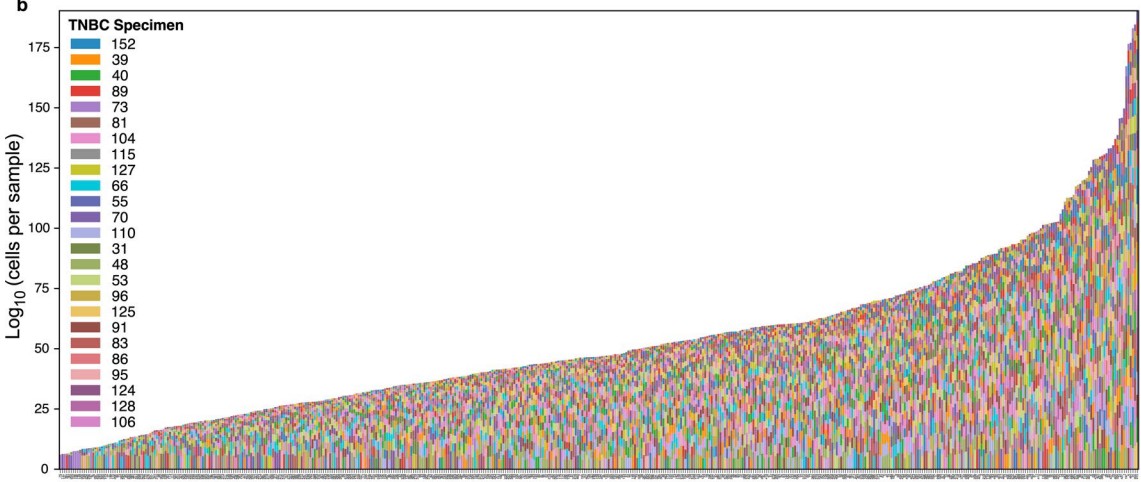

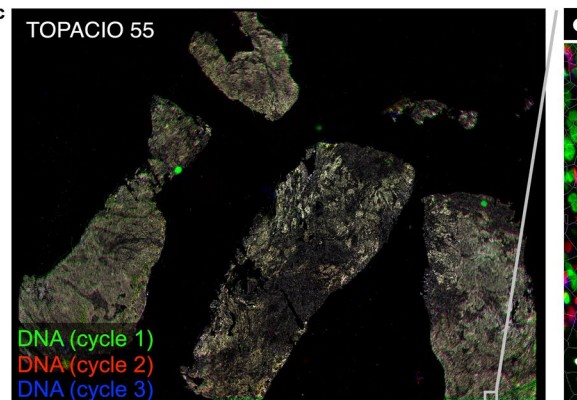

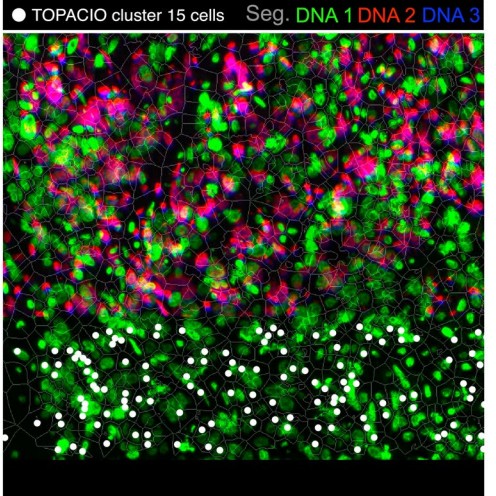

**Extended Data Fig. 3 | Evaluation of pre-QC cell clustering results from Dataset 1 (TOPACIO). a**, Spatial distribution of unclustered (ambiguous) cells from the pre-QC TOPACIO embedding in Fig. 3a as seen in specimen 55 (green scatter points), exhibiting no discernable spatial pattern of sampling bias; Hoechst (gray) shown for tissue context. **b**, Stacked bar charts showing the relative contribution of each patient specimen to pre-QC TOPACIO clusters on log₁₀ scale. **c**, TOPACIO specimen 55 at low (left) and high (right) magnification showing the superimposition of Hoechst signals for the first three imaging cycles: 1 (green), 2 (red), and 3 (blue) demonstrating a cross-cycle image alignment error at the bottom of this image. Box at the bottom-right of the low magnification image shows the location of the higher magnification image. White points in the high magnification image highlight TOPACIO cluster 15 cells formed as a consequence of this image alignment artifact.

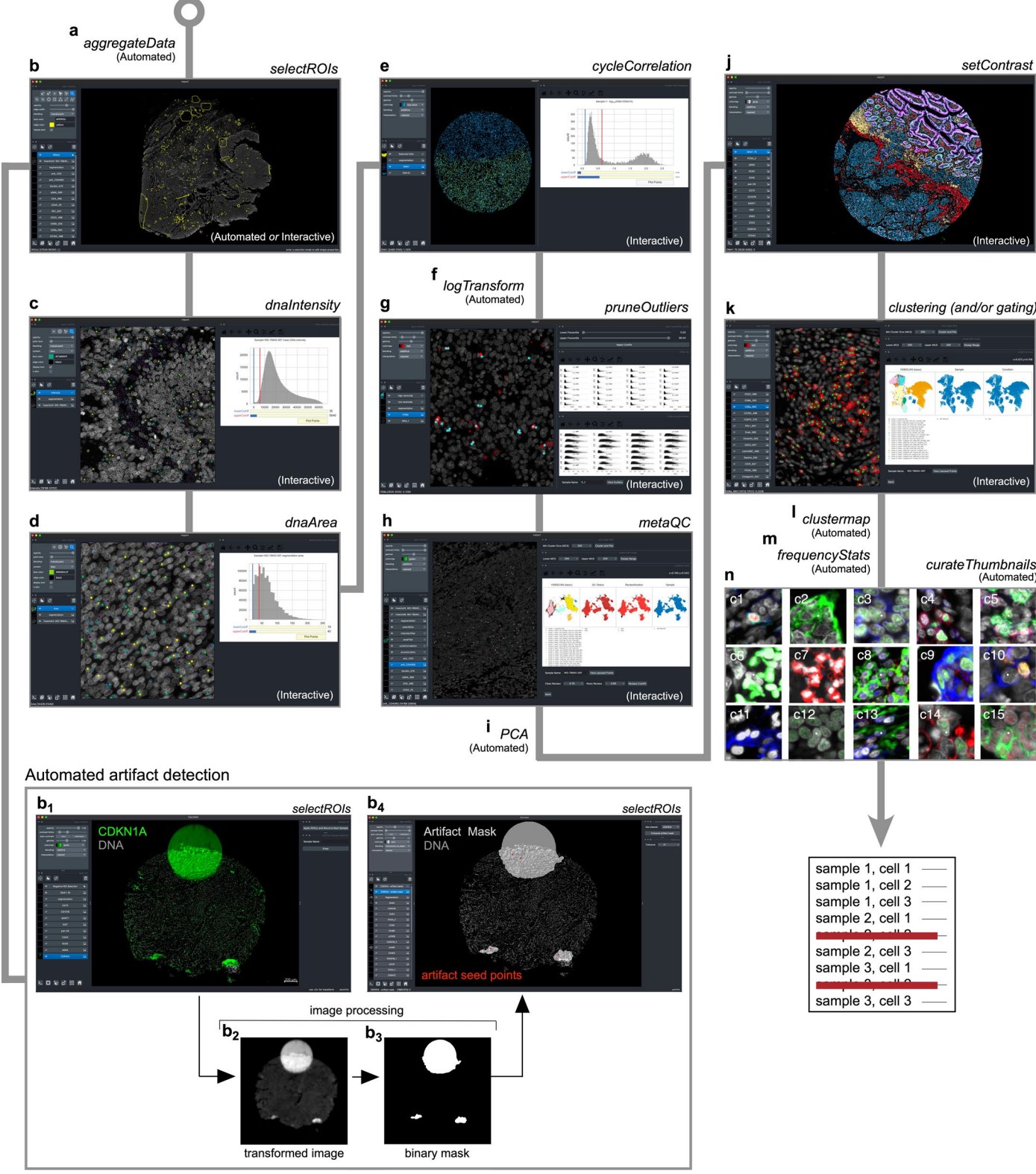

**Extended Data Fig. 4 | See next page for caption.**

**Extended Data Fig. 4 | Identifying and removing noisy single-cell data points with CyLinter.** CyLinter workflow (see project website for implementation details: https://labsyspharm.github.io/cylinter/modules/). **a**, Aggregate data (automated): raw spatial feature tables for all specimens in a batch are merged into a single Pandas (Python) dataframe. **b**, ROI selection (interactive *or* automated): multi-channel images are viewed to identify and gate on regions of tissue affected by microscopy artifacts (negative selection mode) or areas of tissue devoid of artifacts (positive selection mode). **b₁-b₄**, Demonstration of automated artifact detection in CyLinter: **b₁**, CyLinter's *selectROIs* module showing artifacts in the CDKN1A (green) channel of Dataset 3 (EMIT TMA, core 18, mesothelioma). **b₂**, Transformed version of the original CDKN1A image such that artifacts appear as large, bright regions relative to channel intensity variations associated with true signals which are suppressed. **b₃**, Local intensity maxima are identified in the transformed image and a flood fill algorithm is used to create a pixel-level binary mask indicating regions of tissue affected by artifacts. **b₄**, CyLinter's *selectROIs* module showing the binary artifact mask (translucent gray shapes) and their corresponding local maxima (red scatter points) for three artifacts in the image. **c**, DNA intensity filter (interactive): histogram sliders are used to define lower and upper bounds on nuclear counterstain single intensity. Cells between cutoffs are visualized as scatter points at their spatial coordinates in the corresponding tissue for gate confirmation or refinement.
**d**, Segmentation area filter (interactive): histogram sliders are used to define lower and upper bounds on cell segmentation area (pixel counts). Cells between cutoffs are visualized as scatter points at their spatial coordinates in the corresponding tissue for gate confirmation or refinement. **e**, Cross-cycle correlation filter (interactive): applicable to multi-cycle experiments. Histogram sliders are used to define lower and upper bounds on the log-transformed ratio of DNA signals between the first and last imaging cycles ($\log_{10}(DNA_1/DNA_n)$).

Cells between cutoffs are visualized as scatter points at their spatial coordinates in their corresponding tissues for gate confirmation or refinement. **f**, Log transformation (automated): single-cell data are $\log_{10}$-transformed. **g**, Channel outliers filter (interactive): the distribution of cells according to antibody signal intensity is viewed for all specimens as a facet grid of scatter plots (or hexbin plots) against cell area (y-axes). Lower and upper percentile cutoffs are applied to remove outliers. Outliers are visualized as scatter points at their spatial coordinates in their corresponding tissues for gate confirmation or refinement. **h**, MetaQC (interactive): unsupervised clustering methods (UMAP or t-SNE followed by HDBSCAN clustering) are used to correct for gating bias in prior data filtration modules by thresholding on the percent of each cluster composed of clean (maintained) or noisy (redacted) cells. **i**, Principal component analysis (PCA, automated): PCA is performed and Horn's parallel analysis is used to determine the number of PCs associated with non-random variation in the dataset. **j**, Image contrast adjustment (interactive): channel contrast settings are optimized for visualization on reference tissues which are applied to all specimens in the cohort. **k**, Unsupervised clustering (interactive): UMAP (or t-SNE) and HDBSCAN are used to identify unique cell states in a given cohort of tissues. Manual gating can also be performed to identify cell populations. **l**, Compute clustered heatmap (automated): clustered heatmap is generated showing channel *z* scores for identified clusters (or gated populations). **m**, Compute frequency statistics (automated): pairwise t statistics on the frequency of each identified cluster or gated cell population between groups of tissues specified in CyLinter's configuration file (*cylinter_config.yml*) are computed (for example, treated vs. untreated, response vs. no response, etc.). **n**, Evaluate cluster membership (automated): cluster quality is checked by visualizing galleries of example cells drawn at random from each cluster identified in the *clustering* module (k).

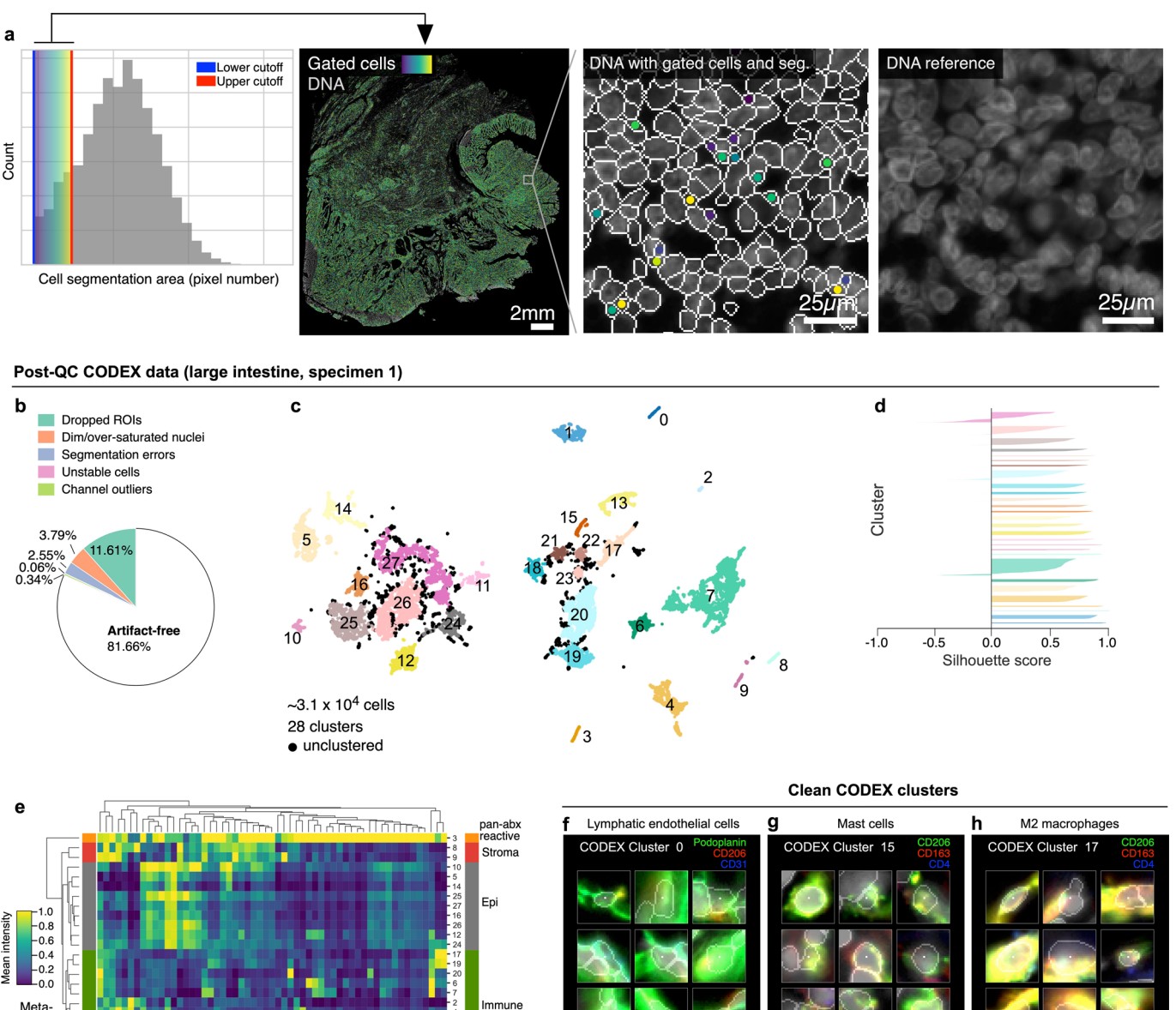

**Extended Data Fig. 5 | Over-segmentation in Dataset 2 (CRC, CyCIF) and Cleaning of Dataset 6 (large intestine, CODEX) with CyLinter. a**, CyLinter-based gating of cells in the CRC image (Dataset 2) according to nuclear segmentation area showing that this image contains several over-segmented nuclei (that is, single nuclei split into multiple segmentation objects). **b**, Fraction of cells in Dataset 6 (large intestine, CODEX, specimen 1) redacted by each QC filter in the CyLinter pipeline. Dropped ROIs, cells dropped by selectROIs module; dim/oversaturated nuclei, cells dropped by dnaIntensity module; segmentation errors, cells dropped by areaFilter module; unstable cells, cells dropped by cycleCorrelation module; channel outliers, cells dropped by pruneOutliers module; artifact-free, cells remaining after QC. **c**, UMAP

embedding of post-QC CODEX clusters showing ~3.1 × 10⁴ cells colored by HDBSCAN cluster. Black scatter points represent ambiguous cells (10.1% of total). **d**, Silhouette scores for post-QC CODEX clusters in (c). **e**, Mean signal intensities for clustering cells in post-QC CODEX data normalized across clusters (row-wise). Five (5) meta-clusters defined by the clustered heatmap dendrogram at the left are highlighted. **f-h**, Top three most highly expressed markers (1: green, 2: red, 3: blue) for clusters 0 (lymphatic endothelial cells, **f**), 15 (mast cells, **g**), and 17 (M2 macrophages, **h**). A single white pixel at the center of each image highlights the reference cell. Nuclear segmentation outlines (translucent outlines) and Hoechst (gray) shown for reference.

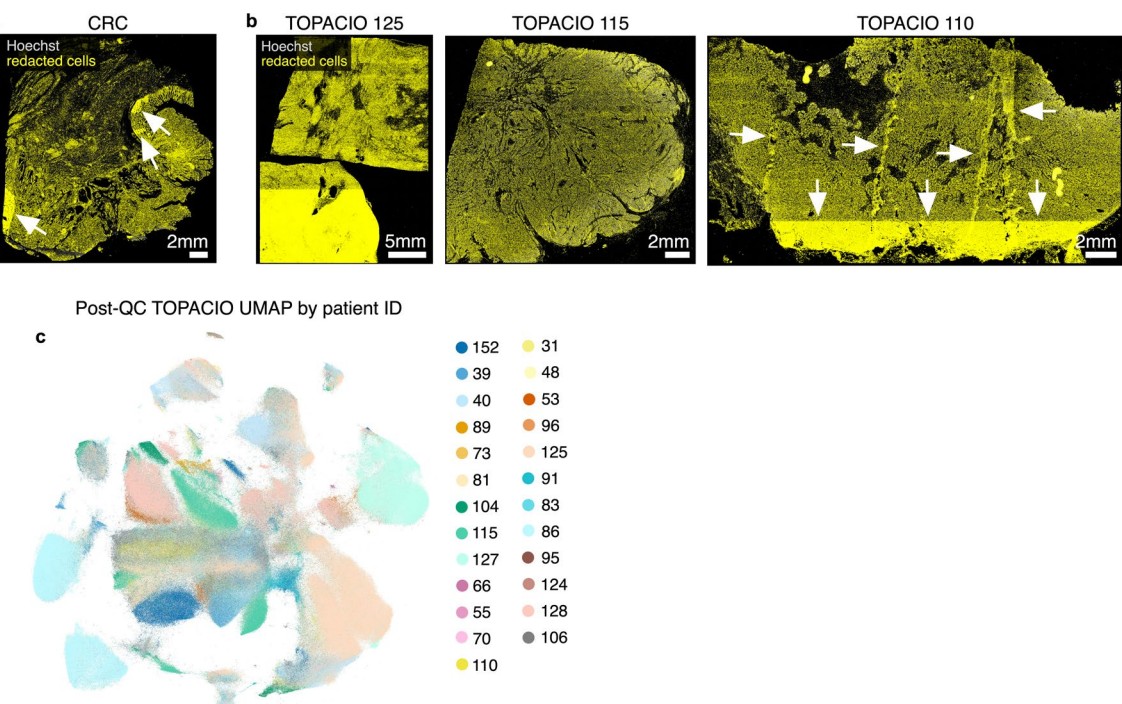

**Extended Data Fig. 6 | Location of cells redacted by CyLinter in Datasets 2 (CRC) and 1 (TOPACIO) and Post-QC TOPACIO UMAP embedding colored by patient ID. a**, **b**: Cells redacted by CyLinter from Dataset 2 (CRC, **a**) and three arbitrary specimens from Dataset 1 (TOPACIO, **b**) demonstrating no discernable bias in the removal of cells from the image with the exception of areas affected by focal artifacts removed using CyLinter's selectROIs module (white arrows). These results are representative of the other 22 tissues in the TOPACIO cohort. **c**, UMAP embedding of post-QC TOPACIO data shown in (Fig. 6b) colored by specimen ID demonstrating patient-specific clustering in tumor cell populations, but not immune and stromal populations (for cluster phenotype identities, refer to Fig. 6b, d, e–i and Online Supplementary Fig. 8).

# Reporting Summary

## Statistics

For all statistical analyses, confirm that the following items are present in the figure legend, table legend, main text, or Methods section.

| n/a | Confirmed | |
|---|---|---|
| ☐ | ☒ | The exact sample size (*n*) for each experimental group/condition, given as a discrete number and unit of measurement |
| ☒ | ☐ | A statement on whether measurements were taken from distinct samples or whether the same sample was measured repeatedly |
| ☐ | ☒ | The statistical test(s) used AND whether they are one- or two-sided *Only common tests should be described solely by name; describe more complex techniques in the Methods section.* |
| ☒ | ☐ | A description of all covariates tested |
| ☒ | ☐ | A description of any assumptions or corrections, such as tests of normality and adjustment for multiple comparisons |
| ☐ | ☒ | A full description of the statistical parameters including central tendency (e.g. means) or other basic estimates (e.g. regression coefficient) AND variation (e.g. standard deviation) or associated estimates of uncertainty (e.g. confidence intervals) |
| ☐ | ☒ | For null hypothesis testing, the test statistic (e.g. *F*, *t*, *r*) with confidence intervals, effect sizes, degrees of freedom and *P* value noted *Give P values as exact values whenever suitable.* |
| ☒ | ☐ | For Bayesian analysis, information on the choice of priors and Markov chain Monte Carlo settings |
| ☒ | ☐ | For hierarchical and complex designs, identification of the appropriate level for tests and full reporting of outcomes |
| ☐ | ☒ | Estimates of effect sizes (e.g. Cohen's *d*, Pearson's *r*), indicating how they were calculated |

*Our web collection on statistics for biologists contains articles on many of the points above.*

## Software and code

Policy information about availability of computer code

| Data collection | Raw microscopy image tiles (RCPNL files) for the datasets described in this study were processed into stitched, registered, and segmented OME-TIFF files using the MCMICRO image-processing software. Corresponding cell x feature CSV files (i.e., spatial feature tables) were also generated by MCMICRO. Specific algorithms implemented in the MCMICRO image processing pipeline are as follows: BaSiC (v1.0.1) — a Fiji/ImageJ plugin for background and shading correction used to perform flatfield and darkfield image correction; ASHLAR (v1.11.1) — a program for seamless mosaic image processing across imaging cycles; Coreograph (v2.2.0) — a program for dearraying TMA corers into individual TIFF and CSV files (https://github.com/HMS-IDAC/UNetCoreograph); UnMICST (v2.4.7) — an implementation of semantic cell segmentation based on the U-Net deep learning architecture; S3segmenter (v1.2.0) — a watershed algorithm used in conjunction with UnMICST (https://github.com/HMS-IDAC/S3segmenter); MCQuant (v1.3.1) — an algorithm used for per cell feature extraction including X,Y spatial coordinates, segmentation areas, mean marker intensities, and nuclear morphology attributes (https://github.com/labsyspharm/quantification). |
|---|---|
| Data analysis | The CyLinter software (v0.0.49) described in this study was written in Python3 (versions 3.8 - 3.11) and continually integrates the latest versions of the following data science libraries: napari, magicgui, pyqt, qtpy, pyyaml, tifffile, scikit-image, zarr, pandas, pyarrow, numpy, matplotlib, seaborn, hdbscan, umap-learn, joblib, scikit-learn, scipy, cellcutter, natsort, numba, svglib, and pypdf2. Figure panels shown in this article were generated using combinations of the individual aforementioned data science libraries and the CyLinter program itself. CyLinter source code is freely available under the MIT open-source license at https://labsyspharm.github.io/cylinter/. Scripts used to generate figure panels in this article may be accessed via a dedicated GitHub repository (https://github.com/labsyspharm/cylinter-paper) archived on Zenodo (https://zenodo.org/records/10067803). |

For manuscripts utilizing custom algorithms or software that are central to the research but not yet described in published literature, software must be made available to editors and reviewers. We strongly encourage code deposition in a community repository (e.g. GitHub). See the Nature Portfolio guidelines for submitting code & software for further information.

## Data

Policy information about availability of data

All manuscripts must include a data availability statement. This statement should provide the following information, where applicable:
- Accession codes, unique identifiers, or web links for publicly available datasets
- A description of any restrictions on data availability
- For clinical datasets or third party data, please ensure that the statement adheres to our policy

This study centers on the analysis of the following seven multiplex imaging datasets: (1) 25 specimens of triple-negative breast cancer from patients enrolled in the TOPACIO clinical trial imaged by CyCIF (ClinicalTrials.gov ID: NCT02657889); (2) a primary human colorectal adenocarcinoma resection imaged by CyCIF (CRC, Lin et al. Cell 2023, PMCID: PMC10019067); (3) a tissue microarray consisting of 123 different healthy and cancerous tissue cores each 1.5 mm in diameter imaged by CyCIF (EMIT TMA22, Synapse: https://www.synapse.org/#!Synapse:syn22345750, Schapiro et al. Nat. Methods 2022, PMCID: PMC8916956); (4) two sections of a single head & neck squamous carcinoma (HNSCC) specimen imaged by CODEX (provided by the laboratory of Kai Wucherpfennig at Dana-Farber Cancer Institute); (5) a whole-slide section of normal human tonsil imaged by mIHC (Synapse: https://www.synapse.org/#!Synapse:syn25174227, Schapiro et al. Nat. Methods 2022, PMCID: PMC8916956); (6) a section of normal human large intestine imaged by CODEX (HuBMAP: https://portal.hubmapconsortium.org/browse/dataset/ae422532f260b3d6fc662aae69b05d33); and (7) a second, independent section of normal human large intestine imaged by CODEX (HuBMAP: https://portal.hubmapconsortium.org/browse/dataset/eaad67a6c6e891ea72cc397c26bd607f). Access to the TOPACIO dataset can be made through the explicit permission of the TOPACIO clinical trial sponsor (Tesaro, Inc.). All other datasets necessary to reproduce the findings in this study can be found at the Sage Bionetworks Synapse data repository at the following URL: https://www.synapse.org/#!Synapse:syn54523217. See Supplementary Table 1 for complete details on dataset identifiers and accession information. All information pertaining to commercial, open-source, or custom code used in the acquisition of previously collected datasets (CRC, EMIT TMA22, HNSCC, tonsil, large intestine) can be found at links to the primary data resources. The TOPACIO dataset was collected during this study using a CyteFinder slide scanning fluorescence microscope and its built-in image acquisition software (RareCyte Inc. Seattle WA).

## Research involving human participants, their data, or biological material

Policy information about studies with human participants or human data. See also policy information about sex, gender (identity/presentation), and sexual orientation and race, ethnicity and racism.

| | |
|---|---|
| Reporting on sex and gender | It is thought that the nature and abundance of microscopy artifacts in images of tissue are not dependent on donor sex or gender. Thus, tissue specimens analyzed in this study were selected without respect to these covariates. Nevertheless, the sex of the donors for the specimens used in this study are as follows: all TOPACIO specimens = female; CRC specimen = male; EMIT TMA22 cores = 44 male/76 female; HNSCC = unknown; tonsil = female; large intestine (samples 1 and 2) = male. |
| Reporting on race, ethnicity, or other socially relevant groupings | The nature and abundance of microscopy artifacts observed in multiplex images of tissue are thought to be independent of race, ethnicity, or social status of the donor. Thus, these factors were not controlled for in our study. Nevertheless, the race of the donors for the specimens used are as follows: CRC, tonsil, and large intestine (sample 2) = Caucasian; large intestine (sample 1) = African American. The race/ethnicity of other tissue donors is not known. |
| Population characteristics | Covariate-relevant population characteristics of the human research participants such as age, genotype, past and current diagnosis, and treatment status are not thought to influence the number and quality of microscopy artifacts in multiplex images of tissue. However, limited demographic information was provided and is as follows: Dataset 1 (TOPACIO) comprises tissue from 25 female patients, Dataset 2 (CRC) is from a 69-year-old white male, Dataset3 (EMIT TMA) comprises cores from 44 males and 76 females between the ages of 21 and 86, Dataset 5 is from a 4-year-old female of European ancestry, Dataset 6 is from a 78-year-old African American male, and Dataset 7 is from a 24-year-old white male. |
| Recruitment | Participants were not recruited for this study. |
| Ethics oversight | The research described in this study complies with all relevant ethical regulations and was reviewed and approved by the Institutional Review Boards (IRBs) at Brigham and Women's Hospital (BWH), Dana-Farber Cancer Institute (DFCI), and Harvard Medical School (HMS). All patient tissue samples were used after informed written consent. |

Note that full information on the approval of the study protocol must also be provided in the manuscript.

# Field-specific reporting

Please select the one below that is the best fit for your research. If you are not sure, read the appropriate sections before making your selection.

☒ Life sciences    ☐ Behavioural & social sciences    ☐ Ecological, evolutionary & environmental sciences

For a reference copy of the document with all sections, see nature.com/documents/nr-reporting-summary-flat.pdf

# Life sciences study design

All studies must disclose on these points even when the disclosure is negative.

| | |
|---|---|
| Sample size | Seven multiplex imaging datasets were used in this study: six whole-slide tissue datasets and a tissue microarray (TMA). Given that this study focuses on the types of artifacts in multiplex tissue imaging and their influence on single-cell data analysis, statistics comparing features within and across populations that would otherwise require computing sample sizes were not performed. However, we believe that the tissue areas |

of the 6 whole-slide Datasets (ranging from 6-353 mm2) and the 123 TMA cores (~2 mm2 each) comprising Dataset 3 together provide sufficient material to robustly survey recurrent artifacts in multiplex imaging data.

**Data exclusions**

Lack of antibody labeling, use of secondary antibodies alone for the purpose of tissue blocking, poor tissue quality during latter imaging cycles, and immunomarker redundancy precluded the use of certain channels in the multiplex images described in this study. The following channels were excluded from each dataset:

TOPACIO: anti-Rat (secondary only), anti-rabbit (secondary only), anti-Goat (secondary only), pSTAT1, Ki67, DNA (cycle 8), STING, pTBK1, pSTAT3, DNA (cycle 9), PCNA, HLAA, and cPARP.

CRC: AF488 (secondary only), AF555 (secondary only), AF647(secondary only), A488 (secondary only), A555 (secondary only), A647 (secondary only), anti_NaKATPase, Ki67_488, Ki67_570.

EMIT TMA22: Rabbit IgG (secondary only), Goat IgG (secondary only), Mouse IgG (secondary only), CD56, CD13, pAUR, CCNE, CDKN2A, PCNA_1, CDKN1B_2.

HNSCC: empty_ch2_cycle1, empty_ch3_cycle1, empty_ch4_cycle1, empty_ch2_cycle4, empty_ch4_cycle4, empty_ch2_cycle5, empty_ch2_cycle6, empty_ch2_cycle8, empty_ch2_cycle9, empty_ch3_cycle9, empty_ch4_cycle9.

Normal tonsil: no channels were excluded from this dataset.

Normal large intestine (sample 1): DRAQ5 (excluded due to its redundancy with Hoechst).

Normal large intestine (sample 2): DRAQ5 (excluded due to its redundancy with Hoechst.

**Replication**

To verify the reproducibility of the findings in this study, we performed in depth analysis of microscopy artifacts and their influence on derived single-cell data across seven different high-plex imaging datasets in the form of whole-slide images and a tissue microarray acquired via one of three different imaging technologies: CyCIF, CODEX, and mIHC. Similar artifacts were observed in all datasets regardless of the technology used to acquire the images and confounded data analysis and interpretation to the same extent.

**Randomization**

Randomization was not performed in this study, as groups of tissues were not compared.

**Blinding**

Blinding was not performed in this retrospective and non-interventional study, as groups of tissues were not compared.

# Reporting for specific materials, systems and methods

We require information from authors about some types of materials, experimental systems and methods used in many studies. Here, indicate whether each material, system or method listed is relevant to your study. If you are not sure if a list item applies to your research, read the appropriate section before selecting a response.

## Materials & experimental systems

| n/a | Involved in the study |
|---|---|
| ☐ | ☒ Antibodies |
| ☒ | ☐ Eukaryotic cell lines |
| ☒ | ☐ Palaeontology and archaeology |
| ☒ | ☐ Animals and other organisms |
| ☒ | ☐ Clinical data |
| ☒ | ☐ Dual use research of concern |
| ☒ | ☐ Plants |

## Methods

| n/a | Involved in the study |
|---|---|
| ☒ | ☐ ChIP-seq |
| ☒ | ☐ Flow cytometry |
| ☒ | ☐ MRI-based neuroimaging |

## Antibodies

**Antibodies used**

The following antibodies were used in the acquisition of the TOPACIO dataset (Name, Clone, Vendor, Catalog number, RRID, dilution):

Donkey anti-Rat A488 (secondary only), polyclonal, Invitrogen, A21208, AB_2535794, 1:1000
Donkey anti-Rabbit A555 (secondary only), polyclonal, Invitrogen, A31572, AB_162543, 1:1000
Donkey anti-Goat A647 (secondary only), polyclonal, Invitrogen, A21447, AB_2535864, 1:1000
CD3 (secondary conjugated), CD3-12, Abcam, ab11089, AB_2889189, 1:200
PD-L1 (secondart conjugated), E1L3N, Cell Signaling Technology, 13684S, AB_2687655, 1:200
53BP1 (secondary conjugated), polyclonal, Bethyl Laboratories, A303-906A, AB_2620256, 1:200
E-Cadherin(A488), 24E10, Cell Signaling Technology, 3199S, AB_2291471, 1:400
panCK(e570), AE1/AE3, EBioscience, 41-9003-82, AB_11218704, 1:800
PD-1(A647), EPR4877(2), abcam, ab201825, AB_2728811, 1:200
CD8a(A488), AMC908, EBioscience, 53-0008-82, AB_2574413, 1:200
CD45(PE), 2D1, R&D, FAB1430P, AB_2237898, 1:100
GrB(A647), 2C5, Santa Cruz, sc-8022AF647, AB_2232723, 1:200
CD163(A488), EPR14643-36, Abcam, ab218293, AB_2889155, 1:400
CD68(PE), D4B9C, Cell Signaling Technology, 79594S, AB_2799935, 1:200

CD20(e660), L26, EBioscience, 50-0202-80, AB_11151691, 1:400
CD4(A488), polyclonal, R&D Systems, FAB8165G, AB_2728839, 1:200
FOXP3(e570), 236A/E7, EBioscience, 41-4777-82, AB_2573609, 1:100
SMA(e660), 1A4, EBioscience, 50-9760-82, AB_2574362, 1:800
CD11b(A488), C67F154, EBioscience, 53-0196-82, AB_2637196, 1:150
pSTAT1(A555), 58D6, Cell Signaling Technology, 8183S, AB_10860600, 1:200
yH2AX(A647), 2F3, Biolegend, 613407, AB_2295046, 1:200
CD57(FITC), NK-1, BD, 561906, AB_395986, 1:100
Ki67(e570), 20Raj1, EBioscience, 41-5699-82, AB_11220278, 1:100
MHCII/HLA-DPB1(A647), EPR11226, Abcam, ab201347, AB_2861375, 1:400
STING(A488), EPR13130, Abcam, ab198950, AB_2889208, 1:400
pTBK1(A555), D52C2, Cell Signaling Technology, 13498S, AB_2943237, 1:200
pSTAT3(A647), D3A7, Cell Signaling Technology, 4324S, AB_10694637, 1:200
PCNA(A488), PC10, Cell Signaling Technology, 8580S, AB_2617115, 1:400
HLA-A(A555), EP1395Y, Abcam, ab207872, AB_2889202, 1:400
cPARP(A647), D64E10, Cell Signaling Technology, 6987S, AB_10699459, 1:100

| Validation | The performance of antibodies used in the collection of the TOPACIO dataset has been previously validated in prior publications (PMIDs: 29993362, 31534232, 30388455) and is indexed on the landing page for recommended CycIF antibodies at https://www.cycif.org/antibodies/recommended. The performance of these antibodies in the current study was confirmed through visual inspection of staining intensity and pattern by board-certified pathologists on tissues known to express their target antigens. Additional validation information on the commercially-available antibodies listed in the "Antibodies Used" subsection above can be found in their associated data product sheets at the manufacture's website. |
| --- | --- |

## Plants

| Seed stocks | *Report on the source of all seed stocks or other plant material used. If applicable, state the seed stock centre and catalogue number. If plant specimens were collected from the field, describe the collection location, date and sampling procedures.* |
| --- | --- |
| Novel plant genotypes | *Describe the methods by which all novel plant genotypes were produced. This includes those generated by transgenic approaches, gene editing, chemical/radiation-based mutagenesis and hybridization. For transgenic lines, describe the transformation method, the number of independent lines analyzed and the generation upon which experiments were performed. For gene-edited lines, describe the editor used, the endogenous sequence targeted for editing, the targeting guide RNA sequence (if applicable) and how the editor was applied.* |
| Authentication | *Describe any authentication procedures for each seed stock used or novel genotype generated. Describe any experiments used to assess the effect of a mutation and, where applicable, how potential secondary effects (e.g. second site T-DNA insertions, mosiacism, off-target gene editing) were examined.* |

