## [Peer Review File · Nature Methods]

Peer Review Information

Manuscript Title: Quality Control for Single Cell Analysis of High-plex Tissue Profiles using CyLinter

Corresponding author name(s): Peter Sorger

Editorial Notes: None

Reviewer Comments & Decisions:

Decision Letter, initial version:

Dear Peter,

Please let me begin by apologizing for our delay in getting these reviews to you, which were slowed down in part by the holidays.

Your Article, "Quality Control for Single Cell Analysis of High-plex Tissue Profiles using CyLinter", has now been seen by two reviewers. As you will see from their comments below, although the reviewers find your work of considerable potential interest, they have raised a number of concerns. We are interested in the possibility of publishing your paper in Nature Methods, but would like to consider your response to these concerns before we reach a final decision on publication.

We therefore invite you to revise your manuscript to address these concerns. Please clarify how much of the workflow is automated, highlight benefits to downstream tasks, and add an demonstration on data from another modality (not CyCIF) while addressing the concerns.

- * include a point-by-point response to the reviewers and to any editorial suggestions
- * please underline/highlight any additions to the text or areas with other significant changes to facilitate review of the revised manuscript
- * address the points listed described below to conform to our open science requirements

* ensure it complies with our general format requirements as set out in our guide to authors at www.nature.com/naturemethods

* resubmit all the necessary files electronically by using the link below to access your home page

[Redacted]

We hope to receive your revised paper within three months. If you cannot send it within this time, please let us know. In this event, we will still be happy to reconsider your paper at a later date so long as nothing similar has been accepted for publication at Nature Methods or published elsewhere.

OPEN SCIENCE REQUIREMENTS

REPORTING SUMMARY AND EDITORIAL POLICY CHECKLISTS

DATA AVAILABILITY

All novel DNA and RNA sequencing data, protein sequences, genetic polymorphisms, linked genotype and phenotype data, gene expression data, macromolecular structures, and proteomics data must be deposited in a publicly accessible database, and accession codes and associated hyperlinks must be provided in the "Data Availability" section.

CODE AVAILABILITY

Please include a "Code Availability" subsection in the Online Methods which details how your custom code is made available. Only in rare cases (where code is not central to the main conclusions of the paper) is the statement "available upon request" allowed (and reasons should be specified).

For more information on our code sharing policy and requirements, please see: <https://www.nature.com/nature-research/editorial-policies/reporting-standards#availability-of->

computer-code

MATERIALS AVAILABILITY

ORCID

Nature Methods is committed to improving transparency in authorship. As part of our efforts in this direction, we are now requesting that all authors identified as 'corresponding author' on published papers create and link their Open Researcher and Contributor Identifier (ORCID) with their account on the Manuscript Tracking System (MTS), prior to acceptance. This applies to primary research papers only. ORCID helps the scientific community achieve unambiguous attribution of all scholarly contributions. You can create and link your ORCID from the home page of the MTS by clicking on 'Modify my Springer Nature account'. For more information please visit please visit www.springernature.com/orcid.

Sincerely,
Rita

Rita Strack, Ph.D.
Senior Editor
Nature Methods

Reviewers' Comments:

Reviewer #1:

Remarks to the Author:

This manuscript from Baker and others is concerned with the broad field of analyzing highly-multiplexed tissue images. This is a large and growing field with multiple technology types impinging on it. The ability to obtain high-dimensional single cell protein (and other non-nucleic acid) data, what

the authors focus on, is highly important, complementary to sequencing, and impactful. This is particularly because of the ability to annotate cell types and their spatial distribution to one another (among other biological reasons), and also for bona-fide single cell analysis in situ. Spatial sequencing often does not have single-cell resolution whereas imaging based approaches can.

There are a variety of substantial challenges for analyzing such datasets on a single cell level. A major one is the difficulty in reliably identifying single cells from the massive amounts of imaging data that is generated. This is particularly challenging for cyclic-based methods where linking cells across images, where each image contains different sets of marker measurements, is essential for data interpretation. In my reading of the manuscript, I understood the authors to be primarily focused on providing a quality control software interface that enables analysts to identify image areas that may generate low quality single cell data and remove them from consideration. More specifically, the authors seem to claim:

1. Imaging artifacts significantly impact single-cell data analysis
2. Development of a quality control software, CyLinter, that facilitates identification of images with artifacts and removes them, or removes problematic sections of images and the single cells therein
3. The software, CyLinter, can salvage otherwise unusable data
4. Their results show that artifact removal should be a standard component of image processing pipelines in spatial tissue profiling

Overall I believe the manuscript is addressing a practically important roadblock to analysis of such datasets, but reads more as a software applications note. There are multiple thoughtful software features and ways of analyzing and looking at the data to cull suspect cells, guided by the deep expertise of such multiplexed image datasets from this lab. However, there is no transformative advance in terms of efficiency that this reviewer could determine---this is software that simply allows humans to interact with the data and manually select regions of interest to censor cells within due to any number of issues. If the purpose of the software is to primarily enable manual ROI selection (Fig. 4) that then updates which single cells should be flagged as problematic, the manuscript's claims may not be fully supported or highly impactful. The authors state more sophisticated machine learning models are being developed to better automatically flag aberrations, but without such features already installed the work's impact is likely limited. There are some analytics provided that can flag certain images or image portions as suspect but a user still must make all these decisions exhaustively it would seem. There is no AI developed to help (although the authors say it is on the agenda but it is challenging---which I'm sure is absolutely the case---these are difficult problems being addressed). Perhaps focus on upstream identification of single cells may be an even bigger yet more impactful challenge.

With regards to the specific claims, I am not sure they are substantially impactful or completely supported by the presented datasets and analyses. The notion that imaging artifacts significantly impact single-cell data analysis seems generally accepted or even obvious, as also would be the idea that artifact removal should be a standard component of image processing in spatial tissue profiling. The quantitative extent to which imaging artifacts is a problem perhaps was generally unknown, but a comprehensive statistical analysis was not seen. More standardization in the artifact removal protocols also would be welcomed and software like CyLinter can contribute. CyLinter helps data analysts remove artifacts but this is done anyways exhaustively going through each image (the authors discuss several days of work still remain either way). They do show how extensive QC of the TAPACIO dataset salvages it for single cell analysis, and this QC was facilitated by CyLinter.

There are five different datasets spanning most current highly-multiplexed imaging technologies highlighted at the beginning of this manuscript. However, it would seem that in practice only two are truly analyzed (CRC and TAPACIO datasets), both using CyCIF primarily developed by the authors. A more comprehensive application set would have been more convincing for generality.

Some more specific comments, and then some additional minor comments, are listed below.

- Fig 1 (and 2ext) shows defect ROIs are mostly localized in a cluster in a UMAP, but how reproducible are the UMAPS from sample to sample and therefore how consistent are such defect signatures across samples?
- Claims of correlation (or not) with tile artifacts and type of biopsy or response to therapy were not quantified or supported (Fig. 3d)
- The enumeration and discussion of artifacts associated with the CRC and TAPACIO datasets is perhaps overkill—that is the sections seem arguably redundant after the presence of artifacts is established
- The data input to CyLinter seems robust and sufficient, and standard (this is a good thing)
- How would the user be confident CyLinter identifies most problematic single cells? If there is not such confidence, use of CyLinter could provide artificial confidence in a dataset and skipping the lengthy but necessary comprehensive manual review process.
- How did the authors determine and the validate over-segmentation affected 16% of the cells in the CRC dataset?
- It is positive that after cell removal, CRC dataset had more cell clusters, but how meaningful were those clusters, and how is that proven, beyond showing six clusters seem to correspond to broad biological categories?

Minor

- Introduction: whole-slide imaging is an FDA requirement for what? And to achieve sufficient statistical power for what?
- Couldn't uneven illumination be fixed with standard flat field or similar normalization adjustments?
- 178: overwhelmingly

Reviewer #2:

Remarks to the Author:

Baker and colleagues address a challenging problem with refreshing honesty and clarity. Personally, it's very encouraging to see careful annotation of commonly encountered artifacts by experts in the field of multiplexed tissue imaging. Beyond documenting many sources of image artifacts, the authors provide a workflow for salvaging single cell data from imperfect but invaluable clinical datasets. They demonstrate the utility of these tools in diverse datasets acquired using distinct imaging modalities: CyCIF, CODEX, and mIHC. The CyLinter software and accompanying site is well documented, clearly presented, and additionally provides example data to allow users to explore CyLinter before applying to their own datasets. This is a timely and welcome addition to the rapidly evolving field of spatial biology. It is original in its conception, thorough in its execution, and flexible in its extension to automated /semi-automated platforms with advances in machine learning. I foresee CyLinter empowering the reuse and refinement of public datasets generated by international consortia. I have a few comments that will strengthen the presentation and interpretation of this well written manuscript.

Major comments:

1. The following changes to the figures would enhance the manuscript:
 - a. Overall: Re-order the panel numbers to be presented in a more logical fashion (left-right, top-bottom). Extended data figure 2, Figure 2, Figure 4, and Supplementary Figure 1 in particular.
 - b. Extended Data Figure 1a: Align and distribute the images more symmetrically.
 - c. Extended Data Figure 1e: What is the "white" pseudo-color? Is that created from an overlap of green and magenta?
 - d. Figure 2 i-m, Figure 3e-g, Figure 5 e-f, and Figure 6d-e are hard to see. Choose 1-2 examples and make bigger?
 - e. Text labels in Extended Data Figure 4 and Extended Data Figure 6 are too small.
 - f. Extended Data Figure 5: Highlight what aspects of workflow are automated an/or require human assistance. What expertise is needed at each step? Please see comment 7 below.
 - g. Extended data Figure 2: Just show one example of coverslip air bubbles and out of focus tissues. Small images cannot see well.
2. Apologies if I missed this point, but it appears from the text and Extended Data Figure 5, that CyLinter is applied exclusively to segmented data. Is it possible to input pixel-level data and cell masks for irregularly shaped cells that are hard to segment (myeloid, stromal, etc)? This might additionally aid with acellular components of the tissue (fibrotic regions, extracellular matrix).
3. Is it possible to perform the image QC (select ROI, adjust DNA intensity, cycle correlation, prune outliers, etc) before cell segmentation? Or will CyLinter not operate without the 4 input files that include data related to cell segmentation? I could see a lot of value performing the QC prior to cell segmentation and/or applying this QC on workflows without segmented data.
4. Does CyLinter output a metadata file documenting the QC modules and thresholds/parameters used for each step? This would be a nice feature to support reproducible workflows and detailed reporting. I could envision a scenario where this file would be uploaded with individual datasets to document how the image data was generated.
5. Is there a threshold where CyLinter fails a dataset? I appreciate the desire to salvage data from critical clinical specimens; however, I think it might be useful to have a "threshold" for data quality that would exclude a dataset. For example, the position ROI selection in Figure 4c is a very unusual shape. How will that impact spatial statistics? I recommend guidelines based on your wealth of imaging experience related to 1) area of tissue masked relative to total area, 2) number of cells lost over cycles, 3) quality/intensity of labeling across tissue specimen, etc.
6. Does CyLinter correct for autofluorescence or do your workflows automatically subtract out background? For example, eosinophils commonly appear "positive" in several channels and create a "noisy cluster" similar to cluster 5 in Extended data Figure 2c. It seems that the same output generated for a tissue fold could aid with this problem.
7. I appreciate that you share the computing requirements for CyLinter in the methods (lines 699-703). Can you also elaborate on the domain expertise required? For example does one have to be an experienced microscopist and/or board-certified histopathologist to utilize CyLinter? Furthermore, can it be readily implemented without a strong computational background?
8. User-introduced bias is discussed and a solution allowing comparison of redacted and unredacted data is shown. How reproducible are workflows generated by different users for the same imaging data? Can you compare image QC between 2 or more investigators? Can you compare/visualize redacted data across individuals?

Minor comments:

1. Lines 210-211: The rationale given for FOXP3, insufficient antibody washing, would presumably

impact the other antibodies used in that cycle. Was that observed or was this very specific to the antibody?

2. What is the origin of the name CyLinter? It's not described in the text.

3. This appears to be very well written and well documented software. Congratulations to the main developer, gjbaker, and colleagues. It is better to structure the installation workflow so that it does not make permanent changes to the users working environment. One minor comment related to the installation of the software is found below. If possible, the binary packages stored in the personal channel (gjbaker) should be moved to the lab channel (labsyspharm) so that they do not depend on a single person's account.

a. `conda update -n base conda`

b. `conda install -n base conda-libmamba-solver`

c. Then use the libmamba solver once: `conda create -n cylinter -c conda-forge -c gjbaker -c labsyspharm python=3 cylinter --solver=libmamba`

Author Rebuttal to Initial comments

NMETH-A54309

Quality Control for Single Cell Analysis of High-plex Tissue Profiles using CyLinter

OVERVIEW OF REVISIONS

We thank the reviewers for their highly constructive critiques of our manuscript. In response, we have thoroughly analyzed a newly available CODEX dataset, added additional automation features to CyLinter (see response to Reviewer 1, Comment 1 below for details), and made a number of additions and modifications to the figure panels and text. We believe that these changes successfully address all comments made by the reviewers and have substantially improved the content, clarity, and potential impact of our work.

A key change is that we added/modified the manuscript text, CyLinter source code, and website documentation to make clear how much of the CyLinter pipeline is automated (see **Extended Data Fig. 4** for an overview). We estimated the time required for CyLinter QC and found that it is but a small fraction of the time required for data collection and downstream, discovery-based analysis (see response to Reviewer 1, Comment 1). We also modified the text and figures to make clear how these existing tools improve single-cell data on both CyCIF and high-quality CODEX data.

At the same time, we appreciate that faster QC would be highly desirable, and we now describe the implementation of Gaussian mixture models (GMMs) that provide smart defaults in three core CyLinter data filtration modules (*intensityFilter*, *areaFilter*, and *cycleCorrelation*). Moreover, in addition to the classical algorithm for automated artifact detection described in our initial submission we have now trained a deep learning (DL) network for automated artifact detection based the Feature Pyramid Network (FPN) architecture (see **Supplementary Note 2** and **Supplementary Fig. 3** in the revised manuscript for details). Our results demonstrate that our strategy for training a DL-based artifact detection model is feasible; however, we currently have insufficient training data to generate a highly accurate and broadly applicable model (current ROC AUC = 0.73). CyLinter is an ideal way to generate such training data. Thus, we have established a deposition site at Sage Synapse for collecting manually curated image artifacts identified with CyLinter (see response to Reviewer 1, Comment 1). Drawing from this database we will further train our DL model to ultimately yield a highly-performant model for integration into future iterations of the CyLinter workflow (alternatively, others can use this freely available data to develop their own methods). Systematically collecting QC data will also make it possible to rigorously address the questions raised by Reviewer 2 about the impact of data redaction on downstream analysis. We also aim to collaborate with NIH consortia HTAN and HuBMAP, which already provide the infrastructure necessary to store and maintain multiplex imaging data, to make QC reports like those generated by CyLinter a metadata standard.

All textual edits describing new data and concepts are highlighted by vertical blue lines at the left margin of our revised manuscript to facilitate re-review. The Open Science reporting summary and editorial checklist have also been updated in accordance with our revisions and we believe that our revised manuscript is in compliance with all other journal format requirements. Lastly, our revision is associated with a new CyLinter release (v0.0.48), which incorporates all of the updated features described in our detailed response to reviewers below.

REVIEWER 1

1) *"In my reading of the manuscript, I understood the authors to be primarily focused on providing a quality control software interface that enables analysts to identify image areas that may generate low quality single cell data and remove them from consideration. More specifically, the authors seem to claim:*

1. *Imaging artifacts significantly impact single-cell data analysis*
2. *Development of a quality control software, CyLinter, that facilitates identification of images with artifacts and removes them, or removes problematic sections of images and the single cells therein*
3. *The software, CyLinter, can salvage otherwise unusable data*
4. *Their results show that artifact removal should be a standard component of image processing pipelines in spatial tissue profiling"*

"However, there is no transformative advance in terms of efficiency that this reviewer could determine---this is software that simply allows humans to interact with the data and manually select regions of interest to censor cells within due to any number of issues. If the purpose of the software is to primarily enable manual ROI selection (Fig. 4) that then updates which single cells should be flagged as problematic, the manuscript's claims may not be fully supported or highly impactful. The authors state more sophisticated machine learning models are being developed to better automatically flag aberrations, but without such features already installed the work's impact is likely limited. There are some analytics provided that can flag certain images or image portions as suspect but a user still must make all these decisions exhaustively it would seem. There is no AI developed to help (although the authors say it is on the agenda but it is challenging---which I'm sure is absolutely the case---these are difficult problems being addressed). Perhaps focus on upstream identification of single cells may be an even bigger yet more impactful challenge"

"With regards to the specific claims, I am not sure they are substantially impactful or completely supported by the presented datasets and analyses."

R1. We thank the reviewer for these comments, which have helped us make the manuscript more compelling. With respect to the reviewer's concerns about innovation, we would like to point out that our work primarily represents a conceptual advance in which we demonstrate the remarkably large impact of common image artifacts on high-dimensional data analysis and single-cell phenotyping. Removing noisy data from an image would seem like an obvious step, but hundreds of spatial profiling studies have been published to date with little to no evidence that systematic QC has been performed. To the best of our knowledge, CyLinter is the first software package for analyzing and removing artifacts from high-plex tissue images stemming from multiple sources at the single-cell level. Using it, the task of processing the 25 images representing the artifact-rich TOPACIO dataset described in our study took a single individual ~5 days to vet for quality, compared to the ~30 days needed to collect the data, and ~6 months to deeply analyze leading to the initial draft of our manuscript.

With respect to the reviewer's concern about automation and efficiency, the revised manuscript clarifies that the suite of 15 QC modules instantiated in the CyLinter software package constitute an efficient system for interactive analysis and elimination of wide-ranging QC-related issues in high-plex images of tissue. This goes well beyond manual ROI selection. Key methods instantiated as semi-automatic or reviewer-supervised automatic modules include: **1)** filtering diffuse out-of-focus and counterstain-oversaturated cells, **2)** removing over- and under-segmented cells according to cell segmentation area, **3)** removing cells that have shifted or fallen off the microscope slide over the course of cyclic imaging studies, **4)** removing cells with excessively bright signals that may have been missed or were too numerous to fully curate via ROIs. Each of these QC strategies systematically identifies cells impacted by specific types of artifacts *without* the requirement of exhaustive ROI selection. Additional CyLinter modules implement unsupervised clustering methods to help guard against inadvertent data selection bias and to allow for the identification and visualization of cell populations flagged as residual dataset noise.

In terms of curation time, ROI selection far outweighs other artifact curation tasks such as filtering out intensity outliers (which is performed through simultaneous percentile thresholding of sample histograms per marker channel). In our initial submission, we described the development and implementation of an algorithm in CyLinter's *selectROIs* module for automated ROI selection (**Extended Data Fig. 5** in the original submission, **Extended Data Fig. 4** in the revised submission). Although it does not use sophisticated machine learning or AI, this algorithm is fast and highly effective at identifying bright anomalies in images (one of the most common artifact types in IF images in our experience). Importantly, the algorithm is also agnostic to the channel in which the artifact resides. ROI masks automatically generated in the *selectROIs* module can also be refined via human review if needed through the tuning of a sensitivity parameter built into the CyLinter user-interface. This makes the *selectROIs* module a convenient and universal detector of bright anomalies such as tissue folds, antibody aggregates, autofluorescence slide debris, and miscellaneous illumination aberrations commonly affecting IF images that conveniently combines automation with human curation and review.

In recognition of the reviewer's concern, we have updated CyLinter to implement automated "smart thresholds" based on Gaussian mixture modeling in the filtration modules described above (1-3) while allowing the user to make final modifications if necessary. This seems to us an appropriate approach, since microscopy is an inherently visual field in which humans remain a gold-standard for quality evaluation. Thus, CyLinter was purposefully developed around a human-in-the-loop design to allow users the ability to closely interact with their data through pixel-level data visualization, image annotation and thresholding, and importantly, confirmation of noisy data selections through visual review of selected points in their corresponding images.

Furthermore, with respect to AI features, we have developed a deep-learning approach to automated artifact curation based on the Feature Pyramid Network (FPN) architecture, a fully convolutional encoder-decoder architecture designed for object detection tasks applicable to semantic image segmentation. Thus far the model has been trained on artifacts identified by three human annotators on 11 serial sections of the CRC specimen described in our initial submission. Our model results demonstrate that the approach works well; however, we currently have insufficient training data to generate a highly accurate model (current ROC AUC = 0.73).

Nevertheless, we believe that CyLinter is an ideal way to generate further training data. Thus, we have established a deposition site at the Synapse data repository (Sage Bionetworks, <https://www.synapse.org/#!Synapse:syn24193163/wiki/624232>) for collecting manually curated image artifacts using CyLinter. We anticipate that further training of our FPN model on such data will ultimately yield sufficient training data to generate a highly-performant model that will be integrated into the CyLinter ecosystem. Systematically collecting QC data will also make it possible to rigorously address the questions raised by Reviewer 2 about the impact of data redaction on downstream analysis.

2) *"The quantitative extent to which imaging artifacts is a problem perhaps was generally unknown, but a comprehensive statistical analysis was not seen."*

R2) We assume that the reviewer is referring to problems with imaging approaches other than CyCIF. In our original manuscript we struggled to identify public non-CyCIF whole-slide datasets that were performed at sufficiently high resolution for CyLinter -type analysis. Fortunately, suitable CODEX datasets have been released via the NIH HubMAP initiative and we have added extensive new analyses of these CODEX samples (human large intestines) to our revised manuscript (see **Supplementary Fig. 1f,g; Fig. 1a,b, j; Extended Data Fig. 1d,e; Extended Data Fig. 2a-d; Extended Data Fig. 5 b-h; Supplementary Fig. 2i,j; and Online Supplementary Figs. 3, 7, 9, 10**)

Other types of spatial data currently available involve the use of lanthanide-tagged antibodies imaged by laser scanners. However, these methods (e.g., imaging mass cytometry—IMC and multiplexed ion beam imaging—MIBI) are substantially lower resolution than optical imaging, generate much smaller images (often containing just a few hundred cells), and more severally impacted by hot pixels, channel spillover, and shot noise^{1–6}. We have nevertheless validated the fundamental compatibility of CyLinter with these types of imaging data (specifically MIBI), as the software program is compatible with any imaging data (OME-TIFF/TIFF file format) and associated single-cell spatial feature tables (CSV).

Lastly, differences in approaches to flatfield correction, tile stitching, and image segmentation among CODEX, CyCIF and MIBI datasets make it challenging (perhaps infeasible) to perform a comprehensive statistical analysis of all types of images and artifacts and their effects on downstream data analysis. Performing such an analysis awaits the incorporation of CyLinter into pipelines such as MCMICRO and the release of resulting data.

References:

- 1: Giesen, C. et al. Highly multiplexed imaging of tumor tissues with subcellular resolution by mass cytometry. *Nat. Methods* 11, 417–422 (2014).
- 2: Baharlou, H., Canete, N. P., Cunningham, A. L., Harman, A. N. & Patrick, E. Mass cytometry imaging for the study of human diseases—applications and data analysis strategies. *Front. Immunol.* 10, 2657 (2019).
- 3: Wang, Y. J. et al. Multiplexed in situ imaging mass cytometry analysis of the human endocrine pancreas and immune system in type 1 diabetes. *Cell Metab.* 29, 769–783 (2019).

4: Wu, M. et al. Single-cell analysis of the human pancreas in type 2 diabetes using multi-spectral imaging mass cytometry. *Cell Rep.* 37, 109919 (2021).

5: Chevrier, S. et al. Compensation of signal spillover in suspension and imaging mass cytometry. *Cell Syst.* 6, 612–620 (2018).

6: Baranski, A. et al. MAUI (MBI analysis user interface)—an image processing pipeline for multiplexed mass-based imaging. *PLoS Comput. Biol.* 17, e1008887 (2021).

3) *“There are five different datasets spanning most current highly-multiplexed imaging technologies highlighted at the beginning of this manuscript. However, it would seem that in practice only two are truly analyzed (CRC and TAPACIO datasets), both using CyCIF primarily developed by the authors. A more comprehensive application set would have been more convincing for generality.”*

R3) As described in our response to query 2 above, our revised manuscript now includes detailed analyses of two additional CODEX datasets of human large intestine that were accessed through the HuBMAP data portal. CODEX is another widely adopted cyclic IF imaging technology that uses oligonucleotide-barcoded antibodies as molecular probes. Our analysis of these additional data show that CODEX images are also impacted by tissue folds, antibody aggregates, and spurious illumination aberrations that lead to artificially bright signals in cells which cause them to form discrete clusters in feature space and hinder the detection of *bona fide* cell populations.

4) *“Fig 1 (and 2ext) shows defect ROIs are mostly localized in a cluster in a UMAP, but how reproducible are the UMAPS from sample to sample and therefore how consistent are such defect signatures across samples?”*

R4) Artifacts stemming from tissue folding, antibody aggregates, illumination aberrations can make affected cells appear brighter than those unaffected by artifacts with biologically-relevant (and often lower) signal intensities. In our experience, this phenomenon is a primary driver of single-cell data variation, with data dimensionality reduction algorithms such as UMAP, t-SNE, and PCA tending to capture it in the first few component axes (e.g., UMAP 1 and UMAP 2). This was shown in our initial manuscript and is now generalized to the additional CODEX datasets shown in our revised manuscript (see **Fig. 1a,b; Fig. 1j; and Extended Data Fig. 1d,e**).

5) *“Claims of correlation (or not) with tile artifacts and type of biopsy or response to therapy were not quantified or supported (Fig. 3d)”*

R5) Quantitative relationships among tile artifacts, biopsy type, and therapeutic response in the TOPACIO dataset have now been statistically analyzed via one-way analysis of variance (ANOVA) followed by pairwise Tukey's Honestly significant difference (HSD) tests to assess differences among groups. The results of these tests are reported on **lines 200-202** of our revised manuscript and—as with all the quantitative results shown in our manuscript—the associated code is provided at the dedicated GitHub repository for this manuscript (<https://github.com/labsyspharm/cylinter-paper>). The results confirm our prior qualitative assessments; namely, that the fraction of tiles with visual artifacts in gross tissue resections is lower than the other two biopsy methods: fine-needle and punch-needle. Whether this reflects a general trend will require more specimens, but we suspect it may hold given our empirical observation that larger tissues (i.e., those with larger

surface area) tend to resist tissue loss and the fractional contribution of individual artifacts to the overall tissue is relatively low. Our statistical analysis also confirms our prior conclusion that the fraction of image tiles affected by artifacts was not associated with therapeutic response to the two-drug combination of niraparib plus pembrolizumab used in the clinical trial.

6) *“How would the user be confident CyLinter identifies most problematic single cells? If there is not such confidence, use of CyLinter could provide artificial confidence in a dataset and skipping the lengthy but necessary comprehensive manual review process.”*

R6) This is an important and central question in the area of QC for high-plex, whole-slide tissue imaging—especially in the case of large tissue cohorts. As far as we can tell, no technology exists that implements this type of QC strategy. We designed CyLinter to strike a balance between automation and human review and there is no reason to believe that it would perform less well than a human working without software support. As the number of cleaned datasets increases, and our AI-based tools for QC automation are further developed, it should be possible to determine whether visual review is always required. At the moment, however, there is little or no evidence in the literature than any comprehensive cell-level QC is being performed on spatial profiling data.

We reiterate that the current CyLinter workflow already incorporates an automated algorithm for flagging bright artifacts in fluorescence images such as antibody aggregates, tissue folds, and spurious other illumination aberrations. It does not, however, account for errors in image processing such as flatfield errors and those in tile alignment and image segmentation. These must ultimately be managed as part of a comprehensive QC workflow. To facilitate development of such workflows, the latest release of CyLinter software release (v0.0.48) generates as an output a single YAML file (*cylinter_report.yml*) containing all the QC metadata (e.g., ROI vertices, histogram thresholds, clustering settings, etc.) necessary to fully document and reproduce a CyLinter analysis starting from raw single-cell feature tables.

As a final thought, artifacts that can only be detected based on morphology are particularly challenging and commonly arise from non-selective antibody staining. Cells affected by this type of artifact can masquerade as *bone fide* cells in clustered (or gated) data. Thus, a comprehensive automated solution to QC for multiplex tissue imaging will also need to account for knowledge of the particular staining patterns of a given antibody panel (which may contain as many as 100 antibodies) to reinforce model accuracy and avoid false negatives (i.e., cells affected by subtle artifacts missed during QC). Our group has already begun looking into solutions to this complex challenge.

7) *“How did the authors determine and the validate over-segmentation affected 16% of the cells in the CRC dataset?”*

R7) Over-segmentation in the CRC dataset was determined using CyLinter's *areaFilter* module. This module allows users to gate on histograms of single-cell data according to cell segmentation area as measured by the number of pixels associated with each segmentation instance which are calculated from the cell segmentation mask for a particular image. With respect to the CRC image, this histogram revealed a shoulder at the lefthand side of the primary peak. Gating on cells within

this shoulder and plotting scatter points at their associated coordinates in the image along with nuclei segmentation outlines revealed that many cell nuclei were split into multiple segmentation instances (i.e., over-segmentation). Our approach to determining over-segmentation in the CRC image is now illustrated in (**Extended Data Fig. 5a**) of our revised manuscript.

8) *"It is positive that after cell removal, CRC dataset had more cell clusters, but how meaningful were those clusters, and how is that proven, beyond showing six clusters seem to correspond to broad biological categories?"*

R8) In our initial submission we provided comprehensive image galleries showing 20 examples of cells from each of the 78 clusters identified in the post-QC CRC dataset (**Online Supplementary Fig. 6**, <https://www.synapse.org/#!Synapse:syn53781719>). This was also done for the pre-QC CRC dataset (**Online Supplementary Fig. 2**, <https://www.synapse.org/#!Synapse:syn53781627>), pre-QC TOPACIO dataset (**Online Supplementary Fig. 4**, <https://www.synapse.org/#!Synapse:syn53782191>), post-QC TOPACIO dataset (**Online Supplementary Fig. 8**, <https://www.synapse.org/#!Synapse:syn53781892>), and has now been extended to a pre- and post-QC CODEX dataset (**Online Supplementary Fig. 3**, <https://www.synapse.org/#!Synapse:syn53781635> and **Online Supplementary Fig. 7**, <https://www.synapse.org/#!Synapse:syn53781730>). Visual review of cells in the post-QC CRC clustering confirm that each exhibited the expected pattern of antibody labeling for biologically-relevant cell states. All other online supplementary data files for our manuscript can be found here: <https://www.synapse.org/#!Synapse:syn24193163/files/>.

9) *"Introduction: whole-slide imaging is an FDA requirement for what? And to achieve sufficient statistical power for what?"*

R9) These statements refer to the associated citation, "*Technical Performance Assessment of Digital Pathology Whole Slide Imaging Devices*"¹ (FDA-2015-D-0230), which itself refers to the requirement for whole-slide images in clinical diagnostics (not small pieces of tissue such as core biopsies). We have also shown previously that it is not possible to compute robust spatial statistics from tissue microarray cores².

References:

- 1:** Health, C. for D. and R. Technical Performance Assessment of Digital Pathology Whole Slide Imaging Devices. *U.S. Food and Drug Administration*. (2019).
- 2:** Lin JR, Wang S, Coy S, Chen YA, Yapp C, Tyler M, Nariya MK, Heiser CN, Lau KS, Santagata S, Sorger PK. Multiplexed 3D atlas of state transitions and immune interaction in colorectal cancer. *Cell*. 2023 Jan 19;186(2):363-381.e19. PMID: PMC10019067

10) *"Couldn't uneven illumination be fixed with standard flat field or similar normalization adjustments?"*

R10) Yes, in many cases uneven field illumination can be corrected. The MCMICRO pipeline for automated processing of multiplex images implements a flat-fielding algorithm called Basic¹ for this purpose. The algorithm works well in most cases, but can fail on datasets in which there are too

few image tiles (~50 or less). We suspect that the limited number of image tiles prevents the algorithm from developing accurate models for flatfield and darkfield fields of view.

Reference:

1: Peng T, Thorn K, Schroeder T, Wang L, Theis FJ, Marr C, Navab N. A BaSiC tool for background and shading correction of optical microscopy images. Nat Commun. 2017 08;8:14836. PMID: PMC5472168

11) "178: overwhelmingly"

R11) This word has been removed in the revised manuscript.

REVIEWER 2

1) *"The following changes to the figures would enhance the manuscript:"*

1a) *"Re-order the panel numbers to be presented in a more logical fashion (left-right, top-bottom). Extended data figure 2, Figure 2, Figure 4, and Supplementary Figure 1 in particular."*

R1a) Figure panels throughout the manuscript have been rearranged to present the data in a consistent orientation from left-to-right, top-to-bottom.

1b) *"Extended Data Figure 1a: Align and distribute the images more symmetrically."*

R1b) The images have been aligned and distributed more symmetrically (see **Supplementary Fig. 1a** of the revised manuscript).

1c) *"Extended Data Figure 1e: What is the "white" pseudo-color? Is that created from an overlap of green and magenta?"*

R1c) We thank the reviewer for their attention to this detail. We have double checked this and confirmed that the white color was derived from the overlap between green (CD20) and magenta (DC-LAMP). On re-evaluation, this raises concern about the specificity of the DC-LAMP antibody applied to this sample, because CD20⁺ B cells are not typically thought to co-express the dendritic cell marker (DC-LAMP). We note that this has no bearing on the results or conclusions of our manuscript, as this mIHC image was not analyzed for cell types, just as an example of the presence of tile stitching artifacts (**Fig. 1k**). To avoid confusion, we have turned off the DC-LAMP channel in this image. The small amount of intense gray signal in the upper left of the updated image (**Supplementary Fig. 1e**) is due to tissue folding, not immunomarker co-expression.

1d) *"Figure 2 i-m, Figure 3e-g, Figure 5 e-f, and Figure 6d-e are hard to see. Choose 1-2 examples and make bigger?"*

R1d) Nine (9) of the original 25 examples are now shown in enlarged format per panel for clarity. Cell image galleries showing 20 examples per cluster for all analyses are also available for review online (<https://www.synapse.org/#!Synapse:syn24193163/files/>).

1e) *"Text labels in Extended Data Figure 4 and Extended Data Figure 6 are too small."*

R1e) Text labels in these figures have been enlarged. Note that some of these panels have been relocated to main figures.

1f) *"Extended Data Figure 5: Highlight what aspects of workflow are automated an/or require human assistance. What expertise is needed at each step? Please see comment 7 below."*

R1f) Which features of the CyLinter pipeline are interactive vs. automated are now articulated in (**Extended Data Fig. 4**) and its accompanying figure legend.

1g) *Extended data Figure 2: Just show one example of coverslip air bubbles and out of focus tissues. Small images cannot see well.*

R1g) Single examples of coverslip air bubbles, out-of-focus tissue, and other artifacts have been enlarged and are shown in (**Fig. 1f** and **Fig. 1g**), respectively, in the revised manuscript. Multiple additional examples of artifacts in the TOPACIO dataset are now also provided for electronic review (**Online Supplementary Fig. 1**, <https://www.synapse.org/#!Synapse:syn53781614>).

2) *“Apologies if I missed this point, but it appears from the text and Extended Data Figure 5, that CyLinter is applied exclusively to segmented data. Is it possible to input pixel-level data and cell masks for irregularly shaped cells that are hard to segment (myeloid, stromal, etc)? This might additionally aid with acellular components of the tissue (fibrotic regions, extracellular matrix).”*

R2) Yes, CyLinter is designed to integrate pixel-level imaging data with tabular data derived from cell masks. The program can take any cell mask regardless of how irregular the object shapes are and allows for co-visualization and analysis with their corresponding multi-channel image files. CyLinter input data files are now described in further detail in the context of (**Fig. 4** and **Extended Data Fig. 4**).

3) *“Is it possible to perform the image QC (select ROI, adjust DNA intensity, cycle correlation, prune outliers, etc) before cell segmentation? Or will CyLinter not operate without the 4 input files that include data related to cell segmentation? I could see a lot of value performing the QC prior to cell segmentation and/or applying this QC on workflows without segmented data.”*

R3) This is an interesting idea that would be relevant to pixel-level learning on images, but is not straightforward for CyLinter to achieve in its current instance, since its workflow was developed around the concept of QC for image-derived, single cell data. Thus, thresholds for DNA intensity and cross-cycle correlation (for example) are currently applied to histograms at the level of cells – it is not clear how they would work in the absence of a segmentation mask and knowledge of what constitutes a cell. As the reviewer notes, removal of bright outlier features and ROI selection is feasible without a segmentation mask and could be added to a future version of the CyLinter program. More generally, we are certainly open to better understanding from the community this use case and how it might inter-operate with cell-level QC for multiplex image analysis.

4) *“Does CyLinter output a metadata file documenting the QC modules and thresholds/parameters used for each step? This would be a nice feature to support reproducible workflows and detailed reporting. I could envision a scenario where this file would be uploaded with individual datasets to document how the image data was generated.”*

R4) This is an excellent suggestion. At the time of initial submission, ROIs and threshold settings were output into individual module-associated output directories along with a graphical overview of the fraction of single-cell data redacted by each module. In the current version of the software program (v0.0.48), which we have released in association with our manuscript revision, these metadata are now combined into a single YAML file (*cylinter_report.yml*). We envision such files becoming a standard in the field when uploading processed imaging datasets to public data portals such as those hosted by HuBMAP ([https://portal.hubmapconsortium.org/search?entity_type\[0\]=Dataset](https://portal.hubmapconsortium.org/search?entity_type[0]=Dataset)) and HTAN (<https://humantumoratlas.org/explore>). Updates to the documentation on the project website have been made to reflect this change.

5) *"Is there a threshold where CyLinter fails a dataset? I appreciate the desire to salvage data from critical clinical specimens; however, I think it might be useful to have a "threshold" for data quality that would exclude a dataset. For example, the position ROI selection in Figure 4c is a very unusual shape. How will that impact spatial statistics? I recommend guidelines based on your wealth of imaging experience related to 1) area of tissue masked relative to total area, 2) number of cells lost over cycles, 3) quality/intensity of labeling across tissue specimen, etc."*

R5) The reviewer makes an excellent point: there are definitely reasons one would not want to include particular data following initial inspection. For example, antibody channels are routinely censored from analysis due to poor immunolabeling performance (i.e., no staining, low signal-to-noise, etc.). A particular concern, addressed in (**Fig. 3** of the revised manuscript detailing noise in the TOPACIO clinical trial dataset), are cases in which downstream analysis would be confounded by QC that disproportionately affected one experimental group – for example, the control arm of a clinical trial. QC that affected specific features of a sample (e.g., immune cells) and altered the statistical analysis of spatial patterns would also be problematic.

We do know that QC issues are diverse and range from too little tissue (i.e., too few cells), to poor antibody quality (lack of staining, off-target labeling) and poor image segmentation, to excessive tissue degradation (at high cycle number) and excessive autofluorescence background. But we feel that we do not have the experience that the reviewer seeks (ours is still a "poverty" rather than "wealth" of experience with QC). The point at which a dataset is judged to fail QC is ultimately informed by the end user's research objectives. Given the scope of the problem, we have added this point to the discussion section of the revised manuscript.

The point raised about spatial statistics of asymmetrical ROIs is one we have also considered. In our experience we find that spatial statistics are best performed on contiguous sections of tissue devoid of tissue tears or other asymmetric tissue geometrics that would lead to inaccurate spatial measurements. Ideally this would mean systematical regions of tissue that would allow radial distances to be accurately computed between all cells. The ROI drawn over the breast cancer tissue now shown in (**Fig. 4g**) of the revised manuscript aims to exclude necrotic regions of the tissue found during initial review to be affected by high antibody background. Irregular ROIs such as these are nevertheless useful in the analysis of cell frequency, so long as they are curated similarly across samples. As a first step in addressing the reviewer's concern, as previously

mentioned in our response to query 4 above, we have modified CyLinter so that it now generates a summary report to help ensure QC transparency and reproducibility.

6) *“Does CyLinter correct for autofluorescence or do your workflows automatically subtract out background? For example, eosinophils commonly appear “positive” in several channels and create a “noisy cluster” similar to cluster 5 in Extended data Figure 2c. It seems that the same output generated for a tissue fold could aid with this problem.”*

R6) This is an excellent point, but CyLinter does not correct for data anomalies (as opposed to those generated by artifacts). In particular, we have found background correction to be surprisingly problematic because it is easy to generate single-cell data with pixel signal intensities at or below zero; and we have shown that the presence of these data points significantly impacts subsequent unsupervised clustering analysis. This was shown in the context of the TOPACIO dataset in our initial submission (see **Supplementary Note 1**). While background subtraction was necessary to improve signal-to-noise in many channels of the TOPACIO data, we nevertheless demonstrated its effect on the qualitative interpretation and quantitative clustering results of this dataset prior to its undergoing QC (see **Fig. 3** and **Supplementary Fig. 2a-d**) which we showed was resolved by CyLinter QC (see **Fig. 6** and **Supplementary Fig. 2e-h**).

The idea of QC for specific cell subtypes (e.g., eosinophils) is one that we intend to add to future version of CyLinter. More generally, as technologies develop, it will be possible to build computational models that can help correct for data anomalies and impute predicted signal intensities across image artifacts. The first step in our lab’s immunofluorescence protocol (CyCIF, <https://www.protocols.io/view/tissue-cyclic-immunofluorescence-t-cycif-version-3-5qpvorbndv4o/v2>) is designed to minimize tissue autofluorescence prior to antibody imaging. An option in our MCMICRO image processing workflow allows users to then implement rolling ball background subtraction (https://scikit-image.org/docs/stable/auto_examples/segmentation/plot_rolling_ball.html) on the stitched and registered mosaic images. However, we do not generally use this procedure because of the aforementioned problem of ≤ 0 pixel intensity values.

7) *“I appreciate that you share the computing requirements for CyLinter in the methods (lines 699-703). Can you also elaborate on the domain expertise required? For example does one have to be an experienced microscopist and/or board-certified histopathologist to utilize CyLinter? Furthermore, can it be readily implemented without a strong computational background?”*

R7) Anyone with sufficient domain knowledge to be comfortable with spatial profiling data can use CyLinter effectively. The program runs on a point-and-click graphical user-interface, so minimal understanding of the common-line is necessary. Stepwise instructions for how to install and execute the program are outlined on the tool’s dedicated website and documentation page (<https://labsyspharm.github.io/cylinter/>). As we continue to develop the CyLinter platform, we also intend to add a browser-based interface to further enhance the tool’s ease of use by scientist with diverse training backgrounds.

The biggest disconnect between what we do in our group compared to general practice (especially in labs new to spatial profiling), lies not in differences in bioinformatic or computational expertise, but in the emphasis we place on visual review of specimens. Our multiplexed data is carefully reviewed by pathologists and cell biologists who often cross-reference adjacent H&E-stained tissue section to aid in making histological annotations. Visual review of multiplexed data (using CyLinter) then follows with iterations of quantitative data analysis and additional visual review. Commonly these tasks are performed by individuals with different expertise, but tissue biology is a skill learnable by any dedicated experimentalist or computational biologist.

8) *User-introduced bias is discussed and a solution allowing comparison of redacted and unredacted data is shown. How reproducible are workflows generated by different users for the same imaging data? Can you compare image QC between 2 or more investigators? Can you compare/visualize redacted data across individuals?*

R8) This is an interesting point. Because CyLinter uses deterministic algorithms, QC by different users will be identical if the thresholds and settings used are identical. As it currently stands, different users can select slightly different values for these parameters, resulting in slightly different sets of redacted data. The rigor of a user's QC also plays into inter-user variability, as a cursory QC is likely to miss artifacts that can impact data analysis and interpretability. CyLinter does not currently implement a module that compares indices of redacted data generated by different users, but we will consider adding such a feature in the future. Ideally, differences in segmentation results and data redaction would feed into an error model that can provide confidence on spatial analysis given uncertainties in the underlying data. This is a long-term vision for our team.

9) *"Lines 210-211: The rationale given for FOXP3, insufficient antibody washing, would presumably impact the other antibodies used in that cycle. Was that observed or was this very specific to the antibody?"*

R9) For reasons that are not fully understood, this artifact was restricted to the FOXP3 (555nm) channel of imaging cycle 5 and appeared as "streaks" of non-specific bright signal across image tiles. Oddly, the other channels in this cycle (CD4, 488nm) and (aSMA, 647nm) were unaffected. Shown below are down-sampled versions of the CD4 (488nm, left), FOXP3 (555nm, center), and aSMA (647nm, right) channels from TOPACIO sample 40 with grids superimposed to demarcate individual image tiles. Images like this were used to estimate the number of tiles affected by optical artifacts. Also shown for reference is a higher resolution image of the same sample (TOPACIO sample 40) as seen in the FOXP3 (green) and DNA (gray) channels without tile grid lines. In many cases we would have simply repeated the experiment on a different section to recover usable FOXP3 data. However, as described in our manuscript, the TOPACIO specimens, like many we have obtained from clinical trials, have only one slide available per patient, making repeat analysis infeasible.

10) *“What is the origin of the name CyLinter? It’s not described in the text.”*

R10). The name CyLinter is a combination of the root work for cell, “cyto-” and the term “linter” used in computer programming to refer to algorithms that flag syntax errors in computer code; this is a nod to the world of computer programming. We have parenthetically added this explanation of the tool’s etymology to the *Software Implementation* subheading of the Methods section in our revised manuscript.

11) *“This appears to be very well written and well documented software. Congratulations to the main developer, gjbaker, and colleagues. It is better to structure the installation workflow so that it does not make permanent changes to the users working environment. One minor comment related to the installation of the software is found below. If possible, the binary packages stored in the personal channel (gjbaker) should be moved to the lab channel (labsyspharm) so that they do not depend on a single person’s account.”*

a. conda update -n base conda

b. conda install -n base conda-libmamba-solver

c. Then use the libmamba solver once: conda create -n cylinter -c conda-forge -c gjbaker -c labsyspharm python=3 cylinter --solver=libmamba

R11) We thank the reviewer for this suggestion. Since the time of initial submission, libmamba has become the default dependency solver for conda; we have updated our documentation page accordingly.

Decision Letter, first revision:

Dear Peter,

Thank you for submitting your revised manuscript "Quality Control for Single Cell Analysis of High-plex Tissue Profiles using CyLinter" (N METH-A54309A). It has now been seen by the original referees and their comments are below. The reviewers find that the paper has improved in revision, and therefore we'll be happy in principle to publish it in Nature Methods, pending minor revisions to comply with our editorial and formatting guidelines.

TRANSPARENT PEER REVIEW

Please note: we allow redactions to authors' rebuttal and reviewer comments in the interest of confidentiality. If you are concerned about the release of confidential data, please let us know specifically what information you would like to have removed. Please note that we cannot incorporate redactions for any other reasons. Reviewer names will be published in the peer review files if the reviewer signed the comments to authors, or if reviewers explicitly agree to release their name. For more information, please refer to our FAQ page.

ORCID

Sincerely,
Rita

Rita Strack, Ph.D.
Senior Editor
Nature Methods

Reviewer #1 (Remarks to the Author):

The authors have done an extensive amount of work to improve the manuscript and I do not have any further recommendations for improvement. I believe CyLinter will be a useful tool for the imaging community.

Reviewer #2 (Remarks to the Author):

Congratulations to all authors! The revised manuscript and updated software package are great.

Final Decision Letter:

Dear Peter,

I am pleased to inform you that your Article, "Quality Control for Single Cell Analysis of High-plex Tissue Profiles using CyLinter", has now been accepted for publication in Nature Methods. The received and accepted dates will be Oct 31, 2023 and May 28, 2024. This note is intended to let you know what to expect from us over the next month or so, and to let you know where to address any further questions.

Over the next few weeks, your paper will be copyedited to ensure that it conforms to Nature Methods style. Once your paper is typeset, you will receive an email with a link to choose the appropriate publishing options for your paper and our Author Services team will be in touch regarding any additional information that may be required. It is extremely important that you let us know now whether you will be difficult to contact over the next month. If this is the case, we ask that you send us the contact information (email, phone and fax) of someone who will be able to check the proofs and deal with any last-minute problems.

Please note that *Nature Methods* is a Transformative Journal (TJ). Authors may publish their research with us through the traditional subscription access route or make their paper immediately open access through payment of an article-processing charge (APC). Authors will not be required to make a final decision about access to their article until it has been accepted. Find out more about Transformative Journals

Authors may need to take specific actions to achieve compliance with funder and institutional open access mandates. If your research is supported by a funder that requires immediate open access (e.g. according to Plan S principles) then you should select the gold OA route,

and we will direct you to the compliant route where possible. For authors selecting the subscription publication route, the journal's standard licensing terms will need to be accepted, including self-archiving policies. Those licensing terms will supersede any other terms that the author or any third party may assert apply to any version of the manuscript.

If you are active on Twitter/X, please e-mail me your and your coauthors' handles so that we may tag you when the paper is published.
